# Efficient Contextual LLM Cascades through Budget-Constrained Policy Learning

**Xuechen Zhang**
University of Michigan
Ann Arbor, MI
zxuechen@umich.edu

**Zijian Huang**
University of Michigan
Ann Arbor, MI
zijianh@umich.edu

**Ege Onur Taga**
University of Michigan
Ann Arbor, MI
egetaga@umich.edu

**Carlee Joe-Wong**
Carnegie Mellon University
Pittsburgh, PA
cjoewong@andrew.cmu.edu

**Samet Oymak**
University of Michigan
Ann Arbor, MI
oymak@umich.edu

**Jiasi Chen**
University of Michigan
Ann Arbor, MI
jiasi@umich.edu

## Abstract

Recent successes in natural language processing have led to the proliferation of large language models (LLMs) by multiple providers. Each LLM offering has different inference accuracy, monetary cost, and latency, and their accuracy further depends on the exact wording of the question (*i.e.*, the specific prompt). At the same time, users often have a limit on monetary budget and latency to answer all their questions, and they do not know which LLMs to choose for each question to meet their accuracy and long term budget requirements. To navigate this rich design space, we propose TREACLE (Thrifty Reasoning via Context-Aware LLM and Prompt Selection), a reinforcement learning policy that jointly selects the model and prompting scheme while respecting the user's monetary cost and latency constraints. TREACLE uses the problem context, including question text embeddings (reflecting the type or difficulty of a query) and the response history (reflecting the consistency of previous responses) to make smart decisions. Our evaluations on standard reasoning datasets (GSM8K, CSQA, and LLC) with various LLMs and prompts show that TREACLE enables cost savings of up to 85% compared to baselines, while maintaining high accuracy. Importantly, it provides the user with the ability to gracefully trade off accuracy for cost.

## 1 Introduction

The success of large language models (LLMs) in recent years has led to a explosion of heterogeneous models and providers, including as Meta's Llama, OpenAI's ChatGPT, and Google's Gemini. As LLMs continue to proliferate in the near future, we envisage a generative AI marketplace with a large variety of providers, LLMs, and deployments. Notably, LLMs have widely varying capabilities and costs: capabilities in terms of accuracy in responding to different types of queries, and cost in terms of monetary price and query latency. As an illustration, the accuracy versus cost tradeoffs of various Llama and GPT LLMs are shown in Figure 1 on grade school math word problems [4]. As can be seen, GPT-3.5 tends to have lower accuracy than GPT-4

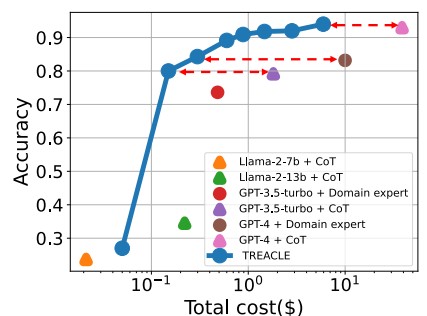

Figure 1: TREACLE chooses LLMs to achieve high accuracy and ~85% cost reduction, compared to individual LLMs.

38th Conference on Neural Information Processing Systems (NeurIPS 2024).

(79% vs 92% respectively), but costs about 20 times less. This heterogeneous array of LLMs can bewilder users who must choose between them.

Another challenge is that the *specific prompt* included in the question plays a critical role in eliciting accurate responses. This is especially true for reasoning problems where prompting a model to explain its reasoning can produce more accurate, but often more costly, answers. Chain-of-thought (CoT) [17] is an example of such a prompting scheme, in which the question includes a few examples of worked out problems, which cost more (due to the additional words included in the question) but also produce more accurate responses. For example, in Figure 1, GPT-4 with CoT (pink triangle) achieves a 92% accuracy, compared to GPT-4 with a domain expert prompt (brown dot, reminding the LLM that it is a "math solver") that achieves 83%. However, using the CoT prompt costs $3.9\times$ more due to the extra words included in the query. A final challenge is that the optimal choice of LLM and prompt depends on the *specific question* being asked; the accuracy of a particular LLM and prompt combination for a particular question is unknown in advance, requiring learning or prediction.

Thus, the heterogeneity of the LLM landscape and the tradeoffs between accuracy and cost make it challenging to determine the optimal strategy of: ***Which LLM to select and how to prompt it, in order to answer all questions while respecting cost constraints?*** To address this, we propose a Thrifty Reasoning via Context-Aware LLM and Prompt Selection (TREACLE) framework. TREACLE is a learning-based approach that solves reasoning questions by automatically selecting which LLM model and prompt to use for each question. Given a cost budget, including a total monetary price across all questions and an average per-query latency, its goal is to maximize the average accuracy of the responses. As shown in Figure 1, TREACLE achieves the Pareto front of individual LLMs by combining them intelligently.

Several recent works utilize multiple LLMs during inference with a cascade design, where queries propagate through a cascade of LLMs, considering the LLMs' accuracy-cost tradeoffs. Most aim to maximize accuracy and lack an explicit way to control long-term costs, as TREACLE has. By posing the problem of LLM and prompt selection as a budget-constrained policy optimization, TREACLE provides a unified approach to efficient LLM cascades (see Table 1). TREACLE makes informed decisions based on the full context of the LLM cascade, including the query embedding, answer statistics, and remaining budget. Overall, this paper makes the following contributions:

- **Characterization of the accuracy, monetary cost, and latency of LLMs.** To understand the trade-offs between the LLMs, we quantify the accuracy and cost of 5 different LLMs (Llama and GPT variants) with 3 different prompt strategies (standard, domain expert, and CoT) on 3 datasets (GSM8K, CSQA, and LLC).
- **An adaptive LLM and prompt selection policy based on reinforcement learning.** TREACLE dynamically chooses the right LLM and prompt for each question. It does this by leveraging context about the current question, re-querying the models if needed to verify the consistency of the responses, and thinking ahead about the remaining budget. We also provide some theoretical justification for TREACLE's key design choices.
- **Extensive evaluations.** We show that TREACLE substantially saves on cost while maintaining high accuracy on mathematical and commonsense reasoning tasks. We demonstrate its robustness to different budgets, question difficulty, price changes, new LLMs, and new unseen task types.

The paper is organized as follows. We describe related work (§2), the problem statement (§3), and our framework (§4). We then describe our experiments (§5) and conclusions (§6).

## 2 Related Work

FrugalGPT [3] is perhaps the closest to this work, as they considered a similar cost-constrained LLM selection problem with a threshold-based policy to select from a sorted list of LLMs. Our approach differs in several key aspects: we utilize a reinforcement learning policy that chooses both LLMs and prompts, rather than a threshold-based scheme; we utilize the full context of the current question to make decisions, including the text embedding of the current question and the history of past responses; and our method can *re-query* the same LLM and aggregate previous responses to estimate the correctness of the current response. Mixture of Thought [20] explored the idea of response consistency in order to choose the right LLMs. The intuition is that higher consistency in the re-queries implies higher confidence in the correctness of the response. TREACLE employs

Table 1: Comparison to related works.

| | Query embedding | Response consistency | Prompt *and* LLM selection | Long-term budget | Robust to new models |
|---|---|---|---|---|---|
| **FrugalGPT** [3] | ✓ | ✗ | ✗ | ✗ | ✗ |
| **AutoMix** [7] | ✓ | ✓ | ✗ | ✗ | ✗ |
| **MoT** [20] | ✗ | ✓ | ✗ | ✗ | ✗ |
| **TREACLE** | ✓ | ✓ | ✓ | ✓ | ✓ |

response consistency as an input feature, along with other features, for LLM selection. AutoMix [7] introduces a "meta-verifier" to estimate whether a response is correct or a more powerful LLM is needed. Both works measure cost as a by-product of combining multiple LLMs rather than long-term constraint across all questions, as we do. Other lines of work include uncertainty estimation or prompt engineering to improve accuracy [6, 18, 20, 13, 1, 9], which is complementary to our work. The related work is summarized in Table 1.

# 3 Problem Statement

We study the natural language query problem of providing correct responses to a series of questions. We focus on reasoning problems (*e.g.*, grade school math problems) because they are challenging with multiple logical steps required to reach a final correct response. The problem involves answering a sequence of $n$ questions $\mathcal{Q}$ with correct responses $\mathcal{Y}$; in other words, we have a set of questions and responses $\{(Q_1, Y_1), (Q_2, Y_2), \ldots, (Q_n, Y_n)\}$. We have a set $\mathcal{M}$ of language models (LLMs) at our disposal, which can be accessed either locally or remotely through APIs: $\mathcal{M} = \{M_1, M_2, \ldots, M_m\}$. Also, we have a choice between $p$ prompt types, $\mathcal{P} = \{P_1, P_2, \ldots, P_p\}$. These models and prompts have different costs (in terms of latency and monetary price) and accuracy. Each "question" can be asked multiple times to the same or different LLMs, which we call a "re-query", in order to possibly obtain a more final accurate response.

The goal is to ensure that as many responses as possible are correct, while simultaneously minimizing the associated costs. This problem can be formulated as a Constrained Markov Decision Process (CMDP), which is represented by a tuple $(\mathcal{Q}, \mathcal{S}, \mathcal{A}, T, r, c, \gamma, B)$, where $\mathcal{Q}$ is the ordered question set; $\mathcal{S}$ is the state space; $T : (\mathcal{Q}, \mathcal{S}) \times \mathcal{A} \times (\mathcal{Q}, \mathcal{S}) \to [0, 1]$ is the transition probability function, *i.e.*, $(Q, s)_{t+1} \sim T(\cdot | (Q, s)_t, a_t)$; $r : (\mathcal{Q}, \mathcal{S}) \times \mathcal{A} \to [R_{\min}, R_{\max}]$ and $c : (\mathcal{Q}, \mathcal{S}) \to \mathbb{R}^+$ denote the reward and cost function; $\gamma \in [0, 1]$ is the discount factor for future reward and cost; and $B$ is the total budget. A policy $\pi : (\mathcal{Q}, \mathcal{S}) \to P(\mathcal{A})$ maps the question-state pairs to a probability distribution over actions. A trajectory $\tau$ is composed of a sequence of (question, state)-action pairs: $\tau = \left\{\tau_{(Q,s)_0}, \tau_{a_0}, \tau_{(Q,s)_1}, \tau_{a_1}, \ldots, \tau_{(Q,s)_L}, \tau_{a_L}\right\}$, where $L$ is the total number of times the LLMs are queried. Note that $L \geq n$, due to the possible re-queries. The cumulative reward and cumulative cost of trajectory $\tau$ are denoted as $R(\tau) = \sum_{t=0}^{L} \gamma^t r(\tau_{(Q,s)_t}, \tau_{a_t})$ and $C(\tau) = \sum_{t=0}^{L} \gamma^t c(\tau_{(Q,s)_t}, \tau_{a_t})$, respectively. The goal of our problem is to learn a policy $\pi$ from $\mathcal{D}$ that maximizes the expected cumulative reward, while satisfying the cumulative cost at the trajectory level:

$$\max_{\pi} \mathbb{E}_{\tau \sim \pi, T}[R(\tau)], \text{ s.t. } \forall \tau \sim \pi, T \quad C(\tau) \leq B. \tag{1}$$

where $\tau \sim \pi, T$ denotes that $\tau$ is generated by executing $\pi$ in $T$. By grouping the rewards and costs by question instead of enumerating all re-queries, the cumulative reward and cost can be re-written as $R(\tau) = \sum_{i=1}^{n} \texttt{reward}(Y_i, \hat{Y}_i)$ and $C(\tau) = \sum_{i=1}^{n} \texttt{cost}(Q_i)$, where $\hat{Y}_i$ is the final response for question $Q_i$, $\texttt{reward}(\cdot)$ is the function that measures the correctness of the final response, $\texttt{cost}(\cdot)$ is the cost function of giving a final response $\hat{Y}_i$ for question $Q_i$. **Cost functions.** We consider two types of costs in this work, monetary price and latency, resulting in two types of cost functions. *(1) Pure monetary price.* LLMs can run remotely, where the monetary price per token is set by the provider (*e.g.*, OpenAI). LLMs can also run locally, where the monetary price depends on a number of factors such as capital expenditures, server cooling and maintenance, electricity, etc. In our setup, the GPT models run remotely and the Llama models, which are free and open-source, run locally. *(2) Monetary price-latency combination.* Monetary price is important for some users (*e.g.*, small companies) while latency plays a more crucial role in other settings (*e.g.*, real-time voice assistants). Users who are latency-sensitive may be willing to pay more for lower latency, whereas others might be more patient and prefer lower prices. TREACLE allows users to choose the trade-off between monetary cost and latency by adjusting a trade-off coefficient $\beta$, where $\texttt{cost} = \texttt{latency} + \beta * \texttt{monetary price}$.

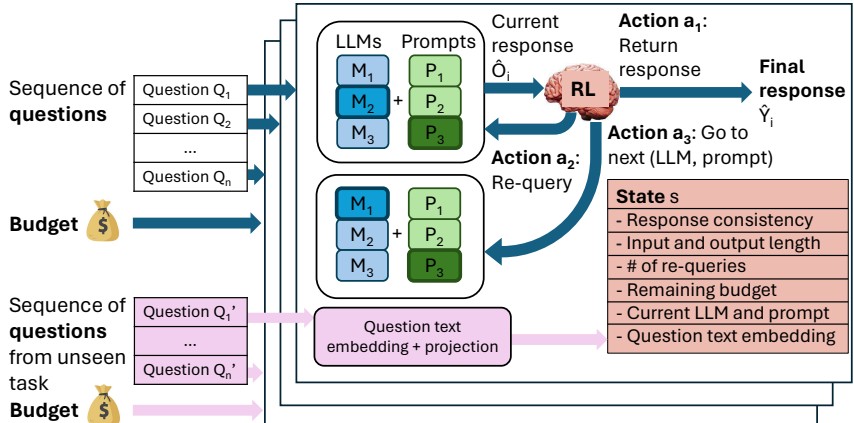

Figure 2: Overview of TREACLE framework. TREACLE decides on the next (LLM, prompt) to query in a context-aware fashion, summarized in the state variable. It can adapt to unseen tasks by projecting the new queries into the text embedding space.

# 4 Proposed Framework: TREACLE

We propose the TREACLE framework, depicted in Figure 2. Let the possible unique combinations of language models and prompts be denoted by $\{(M + P)_1, (M + P)_2, \ldots, (M + P)_K\}$, where $K \leq mp$. When a new question $Q_i$ arrives, TREACLE starts by selecting a model $M \in \mathcal{M}$ and choosing an associated prompt $P \in \mathcal{P}$ to generate a response, denoted as $\hat{O}_i = M(P(Q_i))$. TREACLE returns this as the final response $\hat{Y}_i$ for this question if it has a high degree of confidence in its correctness, and deducts the cost of the question and its response from the total budget $B$. Otherwise, TREACLE can select another LLM $M$ and prompt $P$ (whose choice may be informed by the result of all previously chosen models, prompts, and their responses) and re-query. This iterative process continues until TREACLE returns a final response (based on its learned policy). TREACLE then proceeds to the next question with the remaining budget and repeats the process, until all questions have been answered or there is no remaining budget. We model the problem as a Markov decision process as described in Section 3.

**States.** The state vector contains the following information:

- **Response consistency:** Records all previous responses and the normalized frequency of their occurrences. The intuition is that the consistency of the previous responses can be used as a measure of confidence in the response correctness [16, 7].

- **Input and output length:** The number of tokens in the current query and any preceding responses to the same query. This helps TREACLE understand the monetary price of each query and response, which can differ for each query. It also helps capture the difficulty, as question with longer input or output tend to be harder.

- **Current question's text embedding:** Intuitively, we want to capture the question type or difficulty, which can impact the model and prompt selection decision. TREACLE does this using a text embedding of the query [5].

- **Number of re-queries:** The number of re-queries for each model-prompt pair helps TREACLE decide whether to re-query again or move to the next question.

- **Normalized remaining budget:** Based on the remaining budget, we compute the estimated number of queries for each model prompt pair as follows: $\mathcal{B}_k = \frac{\text{total remaining budget}}{(\text{\# questions remaining})(\text{avg cost per query of } (M + P)_k)}$. The average cost per query is estimated based on the questions seen so far. If there is a large remaining budget, TREACLE may consider re-querying with large models.

**Actions.** The action space $\mathcal{A}$ consists of the following:

- Action $a_1$: Return the current response for $Q_i$ and proceed to the next question $Q_{i+1}$. If no models have been queried yet and this action is chosen, it is equivalent to skipping the question.

- Action $a_2$: Re-query the same model-prompt pair $(M + P)$ for $Q_i$.
- Action $a_3$: Select a new model-prompt pair $(M + P)'$ for $Q_i$.

By allowing re-querying (action $a_2$), the current action influences the next state, by impacting the question under consideration and thus the relevant state features, making this a non-trivial MDP. For $a_3$, we constrained the set of possible model-prompt pairs to a sorted list. In other words, instead of allowing TREACLE to select any possible model and prompting scheme, we sort the $(M + P)_k$ in ascending order of accuracy to cost ratio and only allow TREACLE to select the next element in this list $(M + P)_{k+1}$. The ordering is based on Proposition 1 (discussed below),

**Rewards.** The reward function assigns a positive reward to correct responses. Specifically, $r_{\tau_a}\left(\tau_{(Q,s)}, \tau_{(Q',s')}\right) = \mathbb{P}\left[\hat{Y} = Y | \tau_a = a_1\right] + \lambda \mathbb{P}\left[\hat{O} = Y | \tau_a \in \{a_1, a_2, a_3\}\right]$. For a given question, this combines the accuracy of the final response $\hat{Y}$ with the accuracy of the current response $\hat{O}$ (if there have been re-queries), with a scaling factor $\lambda$ between the two terms. We introduced the second term because without it, we observed that if TREACLE repeatedly chose action $a_2$ (re-querying), this would result in multiple state transitions with 0 reward, until the final response was returned. In other words, including the second term avoids the issue of sparse rewards that resulted from the first term alone. Note that the correct response $Y$ is known only when training TREACLE; during test, the policy executes using the expected reward calculated by the trained policy.

**Design choices and justifications.** We next discuss two key design choices of TREACLE and their theoretical motivation. Proofs are in Appendix B.

*(1) How should the LLMs and prompts be ordered in the cascade?* Recall that action $a_3$ moves to the next $(M + P)$ in the cascade. What is the best ordering of $(M + P)$? Consider the following simplified setting. Suppose each of the $(M + P)$ have a probability of correct response $p_k$ and cost $c_k$. If we had access to an oracle that could tell us when the response of a particular $(M + P)_k$ is incorrect, we could then move on and try the same question with the next option $(M + P)_{k+1}$ in the cascade. We could achieve the highest accuracy using the oracle, and would only have to worry about minimizing the cost to avoid exceeding the budget. In this setting, Proposition 1 below states that the best ordering of the $(M + P)$ options is according to the ratio $\frac{p_k}{c_k}$.

**Proposition 1.** With $K$ (LLM, prompt) options, each with probability of correct answer $p_k$ and cost $c_k$, ordering the options according to their cost-normalized accuracies $\frac{p_k}{c_k}$ minimizes the total cost.

This Proposition motivates TREACLE's ordering of the $(M + P)$ options in the cascade according to their accuracy and cost ratio. This is intuitive: instead of placing the most accurate (LLM, prompt) option early in the cascade, which might incur large cost, we first query LLMs that have high accuracy per unit cost. Note that although the setup of Proposition 1 differs from Equation (1), as the cost is the objective rather than a constraint, the trajectory resulting from the ordering in Proposition 1 is also a solution to Equation (1).

*(2) Do policies that consider response consistency perform well?* Recall that "response consistency" is one of the features in the state vector. We seek to understand the performance of policies that consider this feature; a simple such policy is described in Definition 1 below. It returns a final response to a question if the same response value is repeated $w$ times.

**Definition 1.** For each question $Q_i$, an $w$-consistent policy ($w \geq 2$) sets the final response $\hat{Y}_i = \hat{O}_i$ as soon as $\exists\, \hat{O}_i : \texttt{count}(\hat{O}_i) = w$. If no such $\hat{O}_i$ exists, fall back to $w - 1, w - 2$, etc.

Definition 2 below characterizes how likely the $(M + P)$ are to return an incorrect response. A question can be asked $\Omega$ times to the $(M + P)$ options in the cascade, which may not be unique due to the re-queries.

**Definition 2.** Denote the $\Omega$ LLM-prompt options by $(M+P)_{j=1}^{\Omega}$. Let $\mathbb{P}(M_j(P_j(Q_i)))$ be the output distribution of $(M + P)_j$ on problem $Q_i$. Let $\epsilon := \sum_{i=1}^{n} \sup_{1 \leq j \leq \Omega} \sum_{\hat{O} \neq Y_i} \mathbb{P}(M_j(P_j(Q_i)) = \hat{O})^2$.

With the definitions in hand, we can now lower bound the performance of a 2-consistent policy compared to the optimal learned algorithm in Proposition 2 below. Without loss of generality, we study the case when the reward function is the accuracy.

**Proposition 2.** For the problem stated in Equation (1) that achieves $C_*$, the optimal expected accuracy subject to budget constraints, there exists a 2-consistent policy that achieves an accuracy of at least $C_* - \frac{1}{2}\Omega^2 \epsilon$.

In other words, even a simple policy that allows for re-querying and considers response consistency can achieve close to the optimal reward. This motivates TREACLE inclusion of "response consistency" as a state feature. The proposition applies generally and allows for budgets or text embeddings, and does not require the $(M + P)$ to return accurate responses. Experimentally, we find that our learned RL policy is similar to a 2-consistent policy, as 93.02% of the responses are 2-consistent (GSM8K dataset, \$0.30 budget, $\alpha = \frac{1}{20}$). This suggests that our learned policy may not be far from optimal.

## 5 Experiments

We first describe the experiment setup (§5.1) and then the main results (§5.2). Specifically, we examine robustness to new LLMs and changing API prices (§5.2.1), shifts in question difficulty (§5.2.2), and different reasoning datasets (§5.2.3).

### 5.1 Experiment Setup

We summarize the experiment setup, with full details in Appendix A. We use three representative datasets: **GSM8K [4]**, which contains 8.5K high quality grade school math problems created by human writers; **CSQA [11]**, which consists of 12102 multiple choice commonsense reasoning questions encountered in daily life; and **LLC [17]**, where the task is to concatenate the last letters of words in a name (e.g., "Amy Brown" → "yn"). To evaluate our methods, we perform two steps.

*(1) Collect query-response pairs for (LLM, prompt) combinations*. We collected query-response pairs from each dataset for different combinations of LLM, prompt, and LLM temperature. We used 5 different LLMs: **Llama-2-7b-chat**, **Llama-2-13b-chat** [15], **GPT-3.5-turbo**, **GPT-4**, and **GPT-4-turbo** [10]. These models are of varying sizes (7b, 13b, 154b and 1.76t respectively). The Llama models are open-source and run locally on our servers, while the GPT models rely on commercial APIs. We employ several prompting schemes. A prompt generally consists of two parts: the "content message" containing the question, and the "system message" with additional context.

- The **plain text prompt** submits the questions to the LLM as the content message.
- The **domain expert prompt** feeds information about the question's domain as a system message (*e.g.*, "math solver"), and keeping the user's content message as plain text.
- The **standard few-shot prompt** includes a system message ("Follow the given examples and answer the question" [17]) and the content message, which consists of few-shot examples together with the plain text prompt.
- The **Chain-of-Thought (CoT) few-shot prompt** [17] adds some intermediate explanations to the few-shot examples.

*(2) Train* TREACLE *with the query-response pairs.* We used Deep Q-Network (DQN) [8] to train the reinforcement learning (RL) policy in TREACLE, consisting of a two-layer neural network. For the monetary prices, we use the published per-token prices for the GPT models. Since our local Llama deployments do not have API costs, we set Llama-2-7b's price as $\alpha$ times Llama-2-13b's price, and Llama-2-13b's price as $\alpha$ times GPT-3.5-turbo's price. $\alpha$ varies between $\frac{1}{10}, \frac{1}{20}$ or $\frac{1}{50}$. Our pricing is grounded in reality and similar to actual market rates, as the offered price for Llama is approximately 15% of GPT-3.5-turbo according to current providers [14]. For the latency-accuracy tradeoff, we evaluate different trade-off parameters $\beta = [50k, 500k, 1M]$ in the cost function.

We evaluated the following baseline methods, reproducing the methods as faithfully as possible with a common set of LLMs and prompt options.

- **FrugalGPT** [3]. We reproduce FrugalGPT, which uses a DistilBERT model [12] to estimate the response accuracy. If this estimate is below a threshold, the next LLM in the cascade is queried. This baseline shows how TREACLE compares to the state-of-the-art that lacks re-querying.
- **Calibrated cascade.** We build on FrugalGPT's response accuracy estimation and develop a 2-layer neural network, which takes as input TREACLE's state vector and outputs the estimated response accuracy. If this estimated accuracy is below a threshold, the next LLM in the cascade is queried. This baseline compares TREACLE to a modified FrugalGPT.
- **Majority Voting**. For each query, we output the final response based on the majority vote from $\Omega$ re-queries, based on [16, 20]. We set $N = 2$ based on the best empirical results. The (LLM,

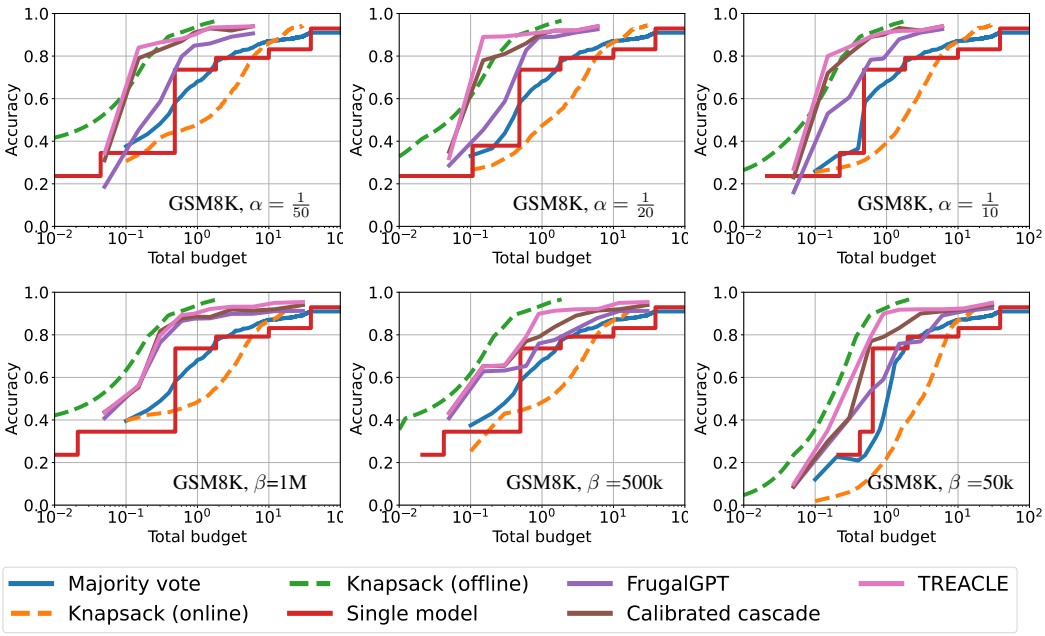

Figure 3: The performance of various methods for different cost functions and budget constraints. The dashed lines are methods that have ground knowledge, which is impractical but illustrates the best achievable performance.

prompt) combinations are progressively queried until their per-question budget runs out. This baseline allows comparison with TREACLE's response consistency feature in the state vector.

- **Offline and online knapsack**. We formulate a multiple choice knapsack problem where the items are the $(M + P)$ combinations. We solve the offline knapsack to find the optimal solution when re-queries are not allowed, and also implement an online version [2]. These baselines show how TREACLE compares to methods with perfect knowledge of question costs and accuracy.

### 5.2 Results

To evaluate the performance of TREACLE, we conduct experiments for different total budgets and cost functions. The results are presented in Figure 3 (additional results on CSQA and LLC are in Appendix C.4.) Across different settings of $\alpha$, $\beta$, and total budget, TREACLE consistently outperforms the baselines and is close to the Offline Knapsack– an approach not feasible in practical deployments. We note that the relatively good performance of the Calibrated Cascade is due to it using the same state vector we designed for TREACLE. We make the following additional observations.

• *Observation 1:* TREACLE *can adapt to different budgets and cost parameters.* All the results in Figure 3, with different budgets and $\alpha, \beta$ parameters were produced after training TREACLE only once. This highlights TREACLE's adaptability to different cost function variations.

• *Observation 2: For limited budgets,* TREACLE *only answers questions that are more likely to produce accurate responses.* For example, for $\beta = 50$k in Figure 3, when the budget is only \$0.05 and insufficient for all queries, 52.7% of the questions TREACLE chooses to answer are correct. For context, the cheapest model (Llama-2-7b) can only answer 23.65% of questions correctly. This suggests TREACLE can evaluate question difficulty and opt not to respond to some questions.

• *Observation 3: For larger budgets,* TREACLE *chooses more powerful (LLM, prompt) combinations.* This is shown in Figure 4a, where as the budget increases, the more powerful models (right side of x-axis) are increasingly selected. Interestingly, we observe that for budgets \$0.3 to \$10, the Llama-2-13b model is queried approximately once per question, despite its suboptimal performance. Even with these larger budgets, it's still beneficial to query Llama before moving onto more powerful models, to see whether its responses are consistent.

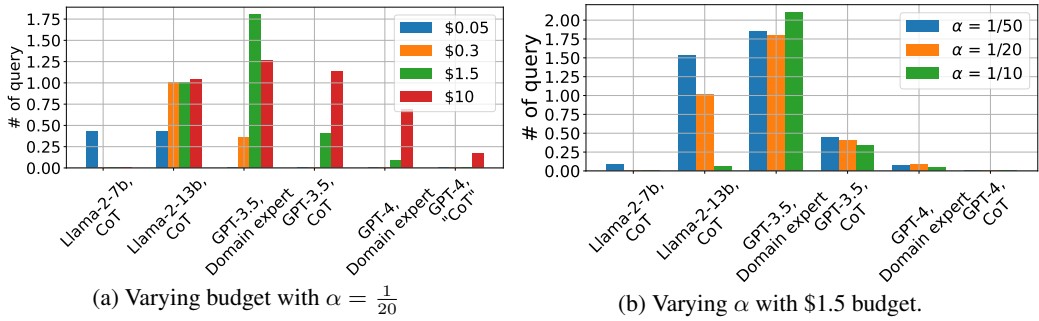

(a) Varying budget with $\alpha = \frac{1}{20}$           (b) Varying $\alpha$ with \$1.5 budget.

Figure 4: Number of times each model is re-queried.

● **Observation 4: Re-querying helps.** We trained both TREACLE and the Calibrated Cascade Algorithm baseline without the ability to re-query. The results are shown in Figure 5, where the dashed line represents method variants that permits re-querying. We observed a notable decrease in accuracy when re-querying was not allowed. Methods without re-querying eventually achieved comparable accuracy with those with re-querying capability, but with significantly larger budgets.

● **Observation 5:** TREACLE*'s choice of model and prompt is impacted by relative LLM prices.* As the relative cost of Llama models decreases ($\alpha$ decreases), TREACLE increasingly utilizes Llama to answer queries, allowing for cost savings, as shown in Figure 4b. This shift enables use

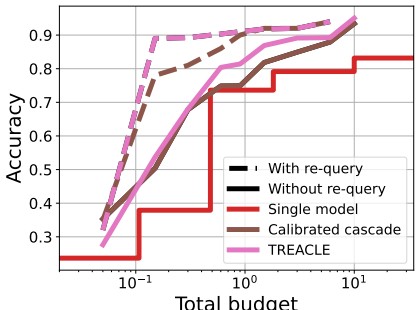

Figure 5: With and without re-querying. $\alpha = \frac{1}{20}$.

of more expensive models like GPT-4 when tackling complex problems, thereby enhancing overall accuracy. When Llama becomes more expensive, TREACLE no longer chooses it. This aligns with our intuition that using Llama to verify response consistency becomes less economical.

### 5.2.1 Addition of new LLMs

LLM development is rapid, with better models continuously emerging, and the API prices set by providers can change at any time. TREACLE's ability to react to such changes is thus an important practical consideration. We show that TREACLE can adapt by fine-tuning itself using few samples. We study two types of LLM updates. (1) *API price adjustment:* In November 2023, OpenAI released GPT-4-turbo, offering performance on par with GPT-4 but at a more affordable price. Concurrently, the price for

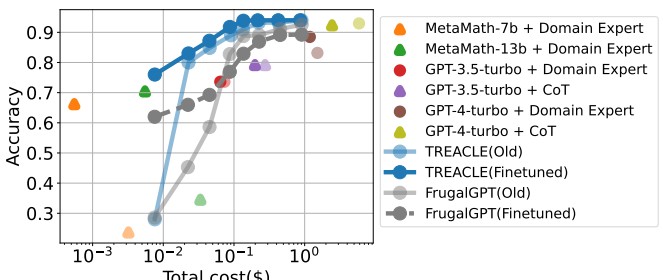

Figure 6: Performance with new LLMs and lowered prices. Lines and dots in light (dark) colors are results with old (new) prices and LLMs. $\alpha = \frac{1}{10}$.

GPT-3.5-turbo was lowered. (2) *Fine-tuned open-source LLMs:* Several domain-specific fine-tuned models with higher accuracy have been released. Specifically, we exchanged Llama-2 for MetaMath [19], which is fine-tuned specifically for GSM8K. For both scenarios, we partitioned the GSM8K test data into 80% validation and 20% test samples, generated new state-action trajectories from the validation set, then fine-tuned TREACLE on these new trajectories. To create a comparable baseline, we also fine-tuned FrugalGPT's DistilBERT.

Firstly, we show the performance of TREACLE with both the API price adjustments and improved LLMs in Figure 6. The individual points on the plot illustrate the changes in the API prices for gpt-3.5-turbo. The lines show the performance of the new TREACLE with new models and prices and the old TREACLE (from previous subsections). The new TREACLE can achieve the peak accuracy with only a \$1 budget, clearly benefiting from the new models and lowered

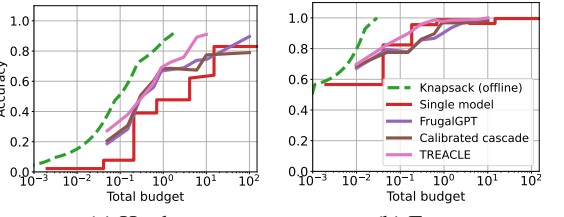

(a) Hard      (b) Easy      (c) Statistics for budget = $1.

|  | # of questions unanswered | | | Budget spent ($) | | |
|---|---|---|---|---|---|---|
|  | All (/1319) | Easy (/500) | Hard (/500) | All | Easy | Hard |
| TREACLE | 1 | 0 | 2 | 0.97 | 0.84 | 0.99 |
| Single Model | 2 | 0 | 0 | 0.69 | 0.69 | 0.70 |
| Cal. Cascade | 2 | 0 | 26 | 0.931 | 0.702 | 0.99 |
| FrugalGPT | 3 | 0 | 31 | 0.96 | 0.713 | 0.99 |

Figure 8: Performance on "easy" and "hard" partitions of the test set. Models are trained on original training data, but must handle a distribution shift in difficulty during test. $\alpha = \frac{1}{20}$.

prices. Benefits are also significant for lower budgets, where the improved TREACLE has significantly higher accuracy, because the lowest performing Llama-2 models were replaced by fine-tuned Metamaths. Finally, for FrugalGPT that relies on a fine-tuned DistilBERT accuracy estimator, performance didn't improve and can even degrade due to distribution shifts and overfitting.

Secondly, in Figure 7 we investigate the sample efficiency of fine-tuning the model with new API prices and LLMs ("Fine-tune" in the figure) compared to training TREACLE from scratch with the new prices and LLMs ("Scratch"). The sample efficiency is important because it can be expensive to collect query-response pairs from new LLMs to further train TREACLE. The results indicate that when there are minor changes to the available LLMs, deploying the previously trained TREACLE can be sufficient.

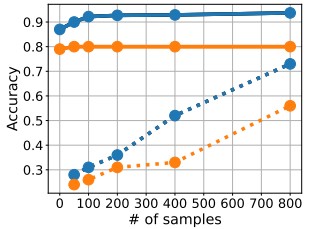

(a) New GPT models and prices (API price adjustment)

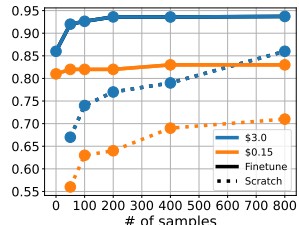

(b) New Llama models (fine-tuned open-source LLMs)

Figure 7: Sample complexity for different budgets with new LLMs. $\alpha = \frac{1}{10}$.

For instance, in Figure 7a when there is limited budget ($0.15) and upgrades to the expensive models, deploying the previously trained TREACLE (# samples = 0) achieves comparable performance to the fine-tuning TREACLE (# samples = 800). On the other hand, when upgrades are introduced to cheaper models (Figure 7b), deploying the old TREACLE may initially result in poor accuracy, but TREACLE can quickly adapt to the new LLM options by fine-tuning with a few number of samples (around 300).

### 5.2.2 Shifts in Question Difficulty

Thus far in the evaluations, easier and harder questions were randomly mixed throughout the training and test sets. In practice, question difficulty may not be uniformly distributed, so we study two types of difficulty distribution shifts: shifts across the training/test sets, and towards the end of the test set.

**Difficulty shifts between training and test.** We divided the GSM8K test set into "hard" and "easy" subsets based on the question difficulty. The difficulty is defined by the number of LLM models correctly answering the questions (more models answering a question correctly roughly means it is easier). Basic performance on the easy and hard questions is shown in Figure 8c. When the questions are hard, each question ends up consuming too much budget, leaving insufficient budget for subsequent questions that then go unanswered. The single model baseline does well in terms of cost and unanswered questions, but has low accuracy. We plot the performance for variable budgets in Figure 8, and find that TREACLE's accuracy remains stable, no matter whether the test distribution shifts to an easier level or a harder level. This is because TREACLE can dynamically adjust based on the remaining budget in online fashion.

**Difficulty shifts within the test set.** To further evaluate the robustness to question difficulty shifts, we test TREACLE with the full test set sorted from easy-to-hard queries or hard-to-easy queries. The hope is that with the help of query text embedding in the state vector (which should capture some estimate of difficulty), TREACLE can remain relatively stable in terms of accuracy even if the ordering of the questions changes. This hypothesis is borne out in Figure 9, while Online Knapsack performs significantly worse than TREACLE if the questions are sorted from hard to easy. This is because much of the budget is wasted on the difficult queries that arrive at the beginning.

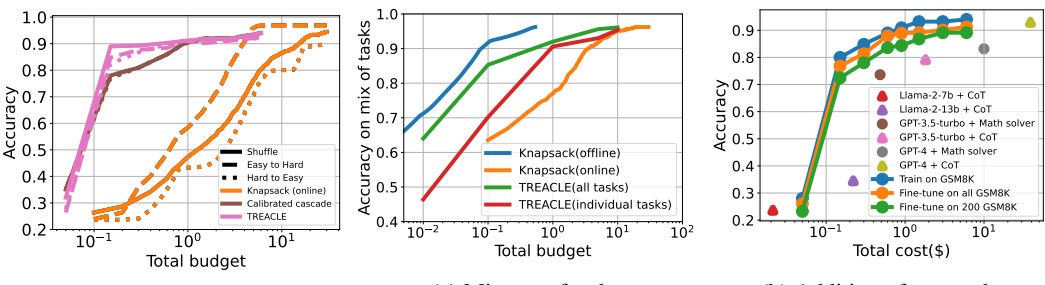

Figure 9: TREACLE is robust to re-ordered question difficulty in the test set. $\alpha = \frac{1}{20}$.

(a) Mixture of tasks       (b) Addition of new task

Figure 10: Performance with different types of reasoning tasks.

### 5.2.3 Different types of reasoning tasks

**Mixture of tasks.** We seek to examine whether one model can handle multiple types of tasks under one common budget (in contrast to the previous experiments with a specialized model for each task). We trained a single model with all 3 datasets and recorded the test accuracy on those datasets. The results shown in Figure 10a for "TREACLE (all tasks)", offline knapsack, and online knapsack are the test accuracy from an equal mix of CSQA, GSM8K and LLC queries. "TREACLE (individual tasks)" is the test accuracy on the same mix of queries, using the models from previous subsections, where each model (corresponding to a task) is assigned to 1/3 of the common budget. "TREACLE (all tasks)" can handle a mixture of tasks under a common budget (*e.g.*, outperforming online knapsack), and can significantly outperform the individual tasks baseline ("TREACLE (individual tasks)") by effectively allocating its common budget across queries of different types.

**New unseen task type.** Consider the scenario where the model has not been trained on certain new tasks. To show that TREACLE can adapt to new tasks easily, we performed additional experiments. The base model is trained using the CSQA dataset, and the unseen new tasks are queries from GSM8K. Interestingly, in our design, we decouple decision making from the task embedding as follows. To transfer from CSQA to GSM8K, we freeze the base RL policy of CSQA (the decision making part), and fine-tune the "text embedding" feature in the state vector (see bottom pink part of Figure 2). How many samples are needed to fine-tune the text embedding for the new task? As shown in Figure 10b, with a budget of 0.6, the original model fully-trained on GSM8K ("train on GSM8K") achieves a test accuracy of 0.848, compared to 0.78 when trained on CSQA and fine-tuned with only 200 additional samples from GSM8K ("fine-tune on 200 GSM8K"). This highlights a relatively small accuracy loss when transferring to new types of unseen tasks. The results suggest that our method can easily adapt to new tasks with only a small amount of additional training.

## 6 Conclusions

We propose TREACLE, a learning-based LLM querying framework that intelligently chooses between LLM and prompt combinations based on question context and past response history. Our experiments show that TREACLE outperforms other baselines and is robust to different budgets, LLM availability and prices, and so on. For future work, we plan to incorporate other features such as privacy into the cost function. We hope our framework can help spur cost-efficient utilization of LLM systems.

**Limitations.** Our work focuses on reasoning problems, and could be extended to generative problems by incorporating new measures of response consistency. The RL policy's budget does not account for the cost of collecting the training data. We plan to freely release the datasets and code so that others can train the same basic policy and adapt that policy to new future LLMs and tasks (as shown through our experiments).

**Broader impact.** This work can make LLMs more accessible to cost-sensitive users.

### Acknowledgements

This work was supported in part by an Adobe Data Science Research award, NSF CCF-2046816, a gift from Google Research, and credits from the Microsoft Accelerating Foundation Models Research grant program. Thank you to Dr. Koyel Mukherjee for helpful discussions and insights on this work.

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

# Appendix

## A Experiment Setup and Implementation Details

### A.1 Datasets

We use three representative datasets for the experiments.

- **GSM8K [4]**: The Grade School Math 8K dataset contains 8.5K high quality grade school math problems created by human writers, in which 7.5K are in the training data and 1K are in the testing data. We further split the 7.5K training data into 6K training data and 1.5K validation data.
- **CSQA [11]**: The Complex Sequential Question Answering dataset consists of 12102 multiple choice commonsense reasoning questions encountered in daily life. The training set, validation set, and testing set contain 9741, 1221 and 1140 samples respectively.
- **LLC [17]** The Last Letter Concatenation task is to concatenate the last letters of words in a name (e.g., "Amy Brown" → "yn").

### A.2 Data collection and training

To evaluate our methods, we perform two steps: (1) Collect query-response pairs for different combinations of LLMs and prompt, then (2) train TREACLE with these pairs.

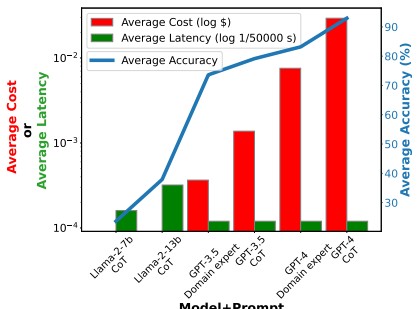

Figure 11: Characterizing accuracy, cost, latency of different model-prompt pairs $(M + P)$ on the GSM8K test dataset. Higher accuracy corresponds to higher price or and lower latency.

**(1) Collecting query-response pairs.** We collect query-response pairs from each dataset for different combinations of LLM, prompt, and LLM temperature. The accuracy, latency, and monetary price of the best combinations are shown in Figure 11, with full results in Table 3 in the Appendix. We selected those combinations according to Proposition 1.

***LLMs.*** We used 5 different LLMs: Llama-2-7b-chat, Llama-2-13b-chat [15], GPT-3.5-turbo, GPT-4, and GPT-4-turbo [10]. These models are of varying sizes (7b, 13b, 154b and 1.76t respectively). The Llama models are open-source and run locally on our servers (one A40 GPU for Llama-2-7b and two A40 for Llama-2-13b), while the GPT models rely on commercial APIs.

***Prompt types.*** We employ several prompting schemes to elicit the reasoning abilities of LLMs. A prompt generally consists of two parts: the "content message" containing the question, and the "system message" with additional context.

- The **plain text prompt** submits the questions to the LLM as the content message (no system message).
- The **domain expert prompt** feeds information about the question's domain as a system message (*e.g.*, "math solver"), and keeping the user's content message as plain text.
- The **standard few-shot prompt** includes a system message ("Follow the given examples and answer the question" [17]) and the content message, which consists of few-shot examples together with the plain text prompt. It tends to improve response accuracy compared to the plain text prompt.

- The **Chain-of-Thought (CoT) few-shot prompt** [17] adds some intermediate explanations to the few-shot examples.

***Temperature.*** The LLM temperature is a configurable parameter that influences the variety of the responses it generates. With a higher temperature, the model may output more diverse but possibly inaccurate responses. We set the temperature to 0 for a new query, and to 0.8 or 1.0 for a re-query for Llama and GPT, respectively.

**(2) Training** TREACLE**.** We used Deep Q-Network (DQN) [8] to train the reinforcement learning (RL) policy in TREACLE, consisting of a two-layer neural network. To generate diverse trajectories consisting of $(s_t, a_t, r_t, s_{t+1})$, we use the collected query-response data and employ $\epsilon$-greedy exploration. For the monetary prices, we use the published per-token prices for the GPT models. Since our local Llama deployments do not have API costs, we set the Llama-2-7b's price as $\alpha$ times Llama-2-13b's price, and Llama-2-13b's price as $\alpha$ times GPT-3.5-turbo's price. $\alpha$ varies between $\frac{1}{10}, \frac{1}{20}$ or $\frac{1}{50}$. Our pricing is grounded in reality and similar to actual market rates, as the offered price for Llama is approximately 15% of GPT-3.5-turbo according to current providers [14]. For the latency-accuracy tradeoff, we evaluate different trade-off parameters $\beta = [50k, 500k, 1M]$ in the cost function. We set the smallest $\beta$ to 50k because then the two terms in the cost function have similar order of magnitude. The details of the cost function values are provided in Table 4 in the Appendix. Unless stated otherwise, we set $\lambda = 5$.

During training, we used the Adam optimizer with a learning rate $1 \times 10^{-4}$, Huber loss as the loss function, and a batch size of 64. Our DQN has three layers with ReLU and softmax activations, and the size of the hidden layer is 128. We set $\lambda = 5$ in the reward function. For re-queries, we set different temperature settings for Llama and GPT (0.8 and 1, respectively) because their ranges are different ($[0, 1]$ and $[0, 2]$ respectively). The actions are selected according to an $\epsilon$-greedy policy. Simply put, the actions are sometimes chosen by the DQN and sometimes sampled uniformly. The probability of choosing a random action starts at $\varepsilon_{\text{START}} = 0.9$ and decays exponentially towards $\varepsilon_{\text{END}} = 0.05$. For the reward decay, we use $\gamma = 0.99$.

### A.3  Baselines

We evaluated the following baseline methods, reproducing the methods as faithfully as possible with a common set of LLMs and prompt options.

- **FrugalGPT** [3]. We reproduce FrugalGPT, which uses a DistilBERT model [12] to estimate the response accuracy. If this estimate is below a threshold, the next LLM in the cascade is queried. This baseline shows how TREACLE compares to the state-of-the-art that lacks re-querying.
- **Calibrated cascade.** We build on FrugalGPT's response accuracy estimation and develop a 2-layer neural network, whose input is a state vector to TREACLE and whose output is the estimated response accuracy. If this estimate is below a threshold (tuned on the validation set), the next LLM in the cascade is queried. This baseline shows how TREACLE compares to an improved version of FrugalGPT.
- **Majority Voting**. For each query, we output the final response based on the majority vote from $N$ re-queries, based on [16, 20]. We set $N = 2$ based on the best empirical results. The (LLM, prompt) combinations are progressively queried until their per-question budget runs out. This baseline allows comparison with TREACLE's response consistency feature in the state vector.
- **Offline and online knapsack**. Given the cost of LLM responses and their accuracy, we formulate a multiple choice knapsack problem where the items are the $(M + P)$ combinations, the values are the correctness probabilities, and the costs are the latency and monetary price functions. Solving this offline knapsack problem gives the optimal solution when re-queries are not allowed. We also implement an online approximation algorithm [2]. These baselines show how TREACLE compares to methods with perfect knowledge of question costs and accuracy.
- **Single model.** The (LLM, prompt) combinations are sorted by increasing cost and accuracy, then the most capable option that fits within the allocated budget is selected for all questions. This baseline shows how TREACLE compares to a fixed single LLM and prompt.

# B Theoretical Justifications for the Cascade Strategy

## B.1 How should the $(M + P)$ be ordered in the cascade?

**Setup 1:** Suppose there are $K$ LLM-prompt pairs each with probability of correct answer $p_i$ and inference cost $c_i$ for $1 \leq i \leq K$. During the cascade, we assume access to an oracle that tells when the answer of a model is incorrect so that we can move to the next model. The procedure continues until a correct answer is obtained. The goal is to minimize the expected cost of inference.

**Lemma 1.** Suppose there are 2 models with probability of correct answers $p_1$ and $p_2$ and inference costs $c_1$ and $c_2$, respectively. Then, the optimal cascade rule is to first query the model with larger cost-normalized accuracy $\frac{p_i}{c_i}$.

*Proof.* The cascade terminates when the first correct answer is obtained. The expected inference cost if we query model 1 first is $C_1 = c_1 p_1 + (c_1 + c_2)(1 - p_1)$, and the expected inference cost if we query model 2 first is $C_2 = c_2 p_2 + (c_1 + c_2)(1 - p_2)$. Note that the accuracy is independent of the cascade order thanks to the oracle. It can be easily shown that $C_1 \leq C_2 \iff \frac{p_1}{c_1} \leq \frac{p_2}{c_2}$. $\square$

**Proposition 1.** With $K$ (LLM, prompt) options, each with probability of correct answer $p_k$ and cost $c_k$, ordering the options according to their cost-normalized accuracies $\frac{p_k}{c_k}$ minimizes the total cost.

*Proof.* Suppose the (LLM, prompt) options are not ordered in terms of $\frac{p_k}{c_k}$. We will prove that swapping the order results in a better cascade. Again recall that the accuracy is independent of the cost because we stop as soon as oracle confirms the answer. The expected accuracy is equal to the probability of at least one (LLM, prompt) option being correct. To proceed, suppose (LLM, prompt) options are not ordered and there is some $\kappa$ such that $\frac{p_\kappa}{c_\kappa} < \frac{p_{\kappa+1}}{c_{\kappa+1}}$. Let us compute the expected change in inference cost when we flip their order.

The change in inference cost arises from the scenarios where (LLM, prompt) option $\kappa$ will be used but $\kappa + 1$ will not be, and vice versa. Define $q_{\kappa-1} = \prod_{i=1}^{\kappa-1}(1 - p_i)$ which is the probability that the first $\kappa - 1$ (LLM, prompt) options fails. The probability of the "excess cost" ($ec$) associated with (LLM, prompt) option $\kappa$ and then $\kappa + 1$ is

$$ec_\kappa = q_\kappa c_\kappa + q_\kappa (1 - p_\kappa) c_{\kappa+1}.$$

This is because if the first $\kappa - 1$ (LLM, prompt) options fail, we definitely pay $c_\kappa$ and only pay $c_{\kappa+1}$ if (LLM, prompt) option $\kappa$ fails. This "excess cost" definition does not reflect the (LLM, prompt) options strictly before $\kappa$ or strictly after $\kappa + 1$. This is because the expected cost of other (LLM, prompt) options arise from the symmetric events with respect to (LLM, prompt) options $\kappa$ and $\kappa + 1$ (either problem is already solved by the time we reach $\kappa$, or both $\kappa$ and $\kappa + 1$ failed).

Conversely, the excess cost associated with $\kappa + 1$ is (*i.e.*, using $\kappa + 1$ first and then $\kappa$)

$$ec_{\kappa+1} = q_\kappa c_{\kappa+1} + q_\kappa (1 - p_{\kappa+1}) c_\kappa.$$

To proceed, observe that if $\frac{p_\kappa}{c_\kappa} < \frac{p_{\kappa+1}}{c_{\kappa+1}}$, then we would be better off by switching the models because

$$ec_{\kappa+1} < ec_\kappa \iff q_\kappa c_{\kappa+1} + q_\kappa (1 - p_{\kappa+1}) c_\kappa < q_\kappa c_\kappa + q_\kappa (1 - p_\kappa) c_{\kappa+1} \tag{2}$$

$$\iff c_{\kappa+1} + (1 - p_{\kappa+1}) c_\kappa < c_\kappa + (1 - p_\kappa) c_{\kappa+1} \tag{3}$$

$$\iff p_\kappa c_{\kappa+1} < p_{\kappa+1} c_\kappa \tag{4}$$

$$\iff p_\kappa / c_\kappa < p_{\kappa+1} / c_{\kappa+1}. \tag{5}$$

$\square$

Experimentally, we verify that our models used TREACLE are sorted according to the decreasing cost-normalized accuracies. The model order (using the purely monetary cost function with $\alpha = \frac{1}{10}$) is shown in Figure 12. With other $\alpha$ values, Llama-2-13b will be cheaper, so the cost-normalized accuracy will always be decreasing.

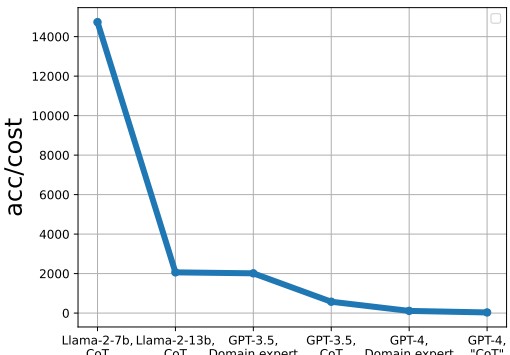

Figure 12: Cost-normalized accuracy

## B.2 Do policies that consider response consistency perform well?

While Proposition 1 provides the optimal ordering rule of LLMs in the cascade, understanding the properties of the optimal trajectory on the cascade is more challenging. To understand the optimal decision rule, we study when a policy terminates for a given question. Given a question, suppose the policy issues up to $\Omega$ queries of the same question to reach a response, where $1 \leq j \leq \Omega$. Experimentally, we have found that the majority of the response statistics are 2-consistent, that is, the policy stops once it observes the same answer twice. Specifically, in our GSM8K experiments with a budget of \$0.30 and $\alpha = \frac{1}{20}$, only 6.98% of the responses achieve consistency greater than 2. This percentage increases modestly to 14.32% when the budget is raised to \$10 (a factor of $33\times$). In this section, we develop a theoretical explanation for this observation by characterizing of the performance of 2-consistent policies and establishing a formal proof of Proposition 2 . The theory relies on Definition 2, which bounds how likely the wrong answers are to be repeated.

**Proposition 2.** For the problem stated in Equation (1) that achieves $C_*$, the optimal expected accuracy subject to budget constraints, there exists a 2-consistent policy that achieves an accuracy of at least $C_* - \frac{1}{2}\Omega^2\epsilon$.

*Proof.* The proof will be achieved via a reduction. Let $\pi_*$ be the optimal policy that achieves an expected accuracy of $C_*$. We now derive a new policy $\pi_2$ from $\pi_*$. Specifically, let $\pi_2$ be same as $\pi_*$ if the trajectory does not attain 2-consistency i.e. the answers are not repeated during the cascade. Otherwise, we let $\pi_2$ terminate early once 2-consistency is achieved for the first time. By construction $\pi_2$ will satisfy the budget constraints as it queries the LLMs strictly less or equal to $\pi_*$. Let $\hat{Y}_i(\pi)$ denote the final (stochastic) answer of a policy $\pi$ given problem $Q_i$. Recall that $Y_i$ denotes the correct answer. Let the random variable $C(\pi_2)$ be the total accuracy of $\pi_2$ summed over all queries where randomness arises from the stochasticity of the LLM responses as well as the policy $\pi_*$. We can lower bound the accuracy of $\pi_2$ as follows

$$\mathbb{E}[C(\pi_2)] = \sum_{i=1}^{n} \mathbb{P}(\hat{Y}_i(\pi_2) = Y_i \mid Q_i) \tag{6}$$

$$\geq \sum_{i=1}^{n} \mathbb{P}(\hat{Y}_i(\pi_2) = Y_i \mid Q_i) - \sum_{i=1}^{n} \mathbb{P}(\hat{Y}_i(\pi_2) \neq \hat{Y}_i(\pi_*) \mid Q_i) \tag{7}$$

$$= C_* - \sum_{i=1}^{n} \mathbb{P}(\hat{Y}_i(\pi_2) \neq \hat{Y}_i(\pi_*) \mid Q_i) \tag{8}$$

$$= C_* - \sum_{i=1}^{n} \mathbb{P}(\text{First 2-consistent answer is wrong} \mid Q_i) \tag{9}$$

$$\geq C_* - \sum_{i=1}^{n} \mathbb{P}(\text{Any wrong 2-consistent answer within } \Omega \text{ responses} \mid Q_i). \tag{10}$$

For example, if the responses are (5, 10, 3, 5, 10) to a given question $\mathbb{P}(\text{First 2-consistent answer is wrong})$ is the probability that "5" is the wrong answer, and

$\mathbb{P}(\text{Wrong 2-consistent answer within }\Omega\text{ responses})$ is the probability that either "5" or "10" is a wrong answer. To proceed, we will upper bound the right hand side of (10). Let $W_{any}$ be the event "Any wrong 2-consistent answer within $\Omega$ responses". Similarly let $W_{uv}$ be the event that the $(M+P)_u$ and $(M+P)_v$ return the same incorrect answer. Note that, for $W_{\text{any}}$ to happen, a $W_{uv}$ event has to happen. Thus, through union bound across $\Omega$ queries given $Q_i$, we have that

$$\mathbb{P}(W_{\text{any}}|Q_i) \leq \sum_{u<v} \mathbb{P}(W_{uv}|Q_i).$$

Plugging in the definition of $W_{uv}$, we obtain

$$\mathbb{P}(W_{\text{any}}|Q_i) \leq \sum_{u<v} \mathbb{P}(W_{uv}|Q_i) \tag{11}$$

$$= \sum_{u<v} \sum_{\hat{O} \neq Y_i} \mathbb{P}(M_u(P_u(Q_i)) = \hat{O})\mathbb{P}(M_v(P_v(Q_i)) = \hat{O}) \tag{12}$$

$$\leq \sum_{u<v} \sum_{\hat{O} \neq Y_i} \frac{\mathbb{P}(M_u(P_u(Q_i)) = \hat{O})^2 + \mathbb{P}(M_v(P_v(Q_i)) = \hat{O})^2}{2} \tag{13}$$

$$\leq \sum_{\hat{O} \neq Y_i} \frac{\Omega^2}{2} \sup_{1 \leq u \leq K} \mathbb{P}(M_u(P_u(Q_i)) = \hat{O})^2. \tag{14}$$

To finalize, we sum over all $n$ questions and use the definition of $\epsilon$ to obtain

$$\sum_{i=1}^{n} \mathbb{P}(W_{\text{any}}|Q_i) \leq \frac{\Omega^2}{2} \sum_{i=1}^{n} \sup_{1 \leq u \leq K} \sum_{\hat{O} \neq Y_i} \mathbb{P}(M_u(P_u(Q_i)) = \hat{O})^2 = \frac{\Omega^2}{2}\epsilon.$$

Plugging the right hand side in equation 10, we conclude with the advertised theorem statement. $\qquad\square$

**Further analysis of 2-consistent policies as a function of incorrect answer likelihoods**. To provide further intuition on Definition 2, let us denote the probability of 1-shot correct answer of a model by $p_{cor} = \mathbb{P}(m_i(\boldsymbol{q}) = a^*)$. Small $p_{cor}$ implies that individual LLM responses are unreliable. However, if $\epsilon$ is relatively small, 2-consistency should return the correct answer with high probability because the majority of the 2-consistent instances will arise from the correct answers. Formalizing this intuition, below we upper-bound the likelihood of error conditioned on 2-consistency as $\mathcal{O}(\epsilon/p_{cor}^2)$.

**Proposition 3.** Recall that $((M+P)_k)_{k=1}^{K}$ are the LLM-prompt pairs used in the cascade. To obtain a more refined bound in terms of the probabilities of correct and wrong answers, let $p_k = \mathbb{P}(M_k(P_k(Q)) = Y)$ for $1 \leq k \leq K$. Also set $p_{\min} = \min_{1 \leq k \leq K} p_k > 0$ and $p_{\max} = \max_{1 \leq k \leq K} p_k$. Let us suppose these hold uniformly over all problem instances $Q$. As $\epsilon \to 0$, over the 2-consistent cascade instances, the probability of error is upper bounded by

$$\text{err}_{\text{upper}} = (1 + o(1))\frac{\sum_{k=2}^{K}(1 - p_{\max})^{-2}(k-1)\mathbf{Q}_k\epsilon}{\sum_{k=2}^{K} \mathbf{P}_k}. \tag{15}$$

Here $\mathbf{P}_k$ is the likelihood that 2-consistent correct answer will first be achieved at the $k$'th model and $Q_k$ is the likelihood that first $k-1$ models fail to achieve 2-consistency. $o(1)$ term captures the higher order terms that arise from the probability of the 2-consistent incorrect cascades (which is vanishing compared to $\mathbf{P}_k, \mathbf{Q}_k$).

*Proof.* In equation 15, the "$(1 - p_{\max})^{-2}\epsilon(k-1)\mathbf{Q}_k$" term in the numerator upper bounds the likelihood that, an incorrect 2-consistent answer will be generated precisely at the $k$th model under 2.

Denote the probability of the event "correct answer never appears until model $k-1$" by $F_k = \prod_{j<k}(1 - p_j)$. As $\epsilon \to 0$, for $2 \leq k \leq K$, these take the simplified closed forms

$$\mathbf{P}_k = p_k F_k \sum_{j<k} \frac{p_j}{1 - p_j} = p_k \sum_{j<k} p_j \prod_{l<k, l\neq j}(1 - p_l) \tag{16}$$

$$\mathbf{Q}_k = F_k(1 + \sum_{j<k} \frac{p_j}{1 - p_j}) = \sum_{j\leq k} p_j \prod_{l<k, l\neq j}(1 - p_l) + \prod_{j<k}(1 - p_j). \tag{17}$$

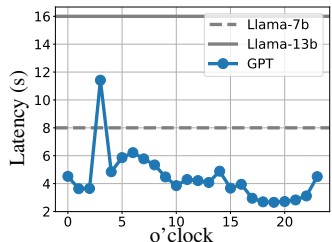

Figure 13: End-to-end latency of querying LLM models over a 24-hour period.

Table 2: Performance with time-varying API query latency.

| Method | Accuracy with Time-varying Latency | Accuracy assuming Constant Latency | Update time (s) |
|---|---|---|---|
| TREACLE | 86.4 | 76.1 | 0.02 |
| Calibrated Cascade | 80.1 | 75.7 | 617.30 |

For a fixed $k$, let us set the ratio

$$\text{rat}_k = \frac{\mathbf{Q}_k}{\mathbf{P}_k} \leq \frac{1 + \sum_{j<k} \frac{p_j}{1-p_j}}{p_k \sum_{j<k} \frac{p_j}{1-p_j}} = \frac{1+M}{p_k M} \tag{18}$$

where $M = \sum_{j<k} \frac{p_j}{1-p_j}$. If $M \geq 1$, we have that $\text{rat}_k \leq \frac{2M}{p_k M} \leq \frac{2}{p_k}$. If $M \leq 1$, we have that $\text{rat}_k \leq \frac{2}{p_k M} \leq \frac{2}{p_k \sum_{j<k} p_j}$. Thus, we obtain

$$\frac{\mathbf{Q}_k}{\mathbf{P}_k} \leq \frac{2}{p_k \cdot \min(1, \sum_{j<k} p_j)} \tag{19}$$

This in turn implies that the error is controlled by the worst case ratio $(k-1)\mathbf{Q}_k/\mathbf{P}_k$ given by $\max_{k \leq K} \frac{2(k-1)}{p_k \cdot \min(1, \sum_{j<k} p_j)}$. Using $p_k \cdot \min(1, \sum_{j<k} p_j) \geq \min((k-1)p_{\min}^2, p_{\min}) \geq p_{\min}^2$, we obtain the conclusion that

$$\text{err}_{\text{upper}} \leq (2 + o(1))\frac{(1-p_{\max})^{-2}\epsilon}{\min(p_{\min}^2, p_{\min}/(K-1))} \leq (2 + o(1))\frac{(K-1)\epsilon}{(1-p_{\max})^2 p_{\min}^2} \propto \frac{(K-1)\epsilon}{p_{\min}^2} \tag{20}$$

$\square$

# C   Additional Results

## C.1   Time-Varying API Query Latency

The latency of querying LLM APIs (such as OpenAI's GPT models) may vary over the short term time based on network congestion or LLM inference time. To showcase this, we recorded traces of the API latency (including communication and communication latency) over a 24-hour period. The measurements are shown in Figure 13. We also modified TREACLE and the experimental setup slightly. Each hour, TREACLE attempts to answer the entire GSM8K test set with a budget of $0.6, using the historical average latency from the previous hour to update the per-query cost in the denominator of $\mathcal{B}_k$. The budget resets every hour. We compare this to the vanilla TREACLE shown previously, which has the same total budget and uses a fixed latency in the denominator of $\mathcal{B}_k$ for the entire 24-h period; for example, for GPT models, it assumes a fixed average latency of 6 s. The results in Table 2 indicate that TREACLE that adapts to time-varying achieves higher accuracy than the version that assumes constant latency. The update time to calculate the new $\mathcal{B}_k$ each hour is also minimal, at 20 ms.

## C.2   Model and Prompt Characterization

We show the model details, prompt strategies, temperature, and various details of each configuration in Table 3 Note that these are the model-prompt combinations chosen in our framework because of the evaluation in Table 4.

Table 3: Characterization of LLM performance in terms of accuracy, latency, and price, for a single query with temperature equal to 0. The Llama models do not have a direct monetary price because they are open-source and we run them locally.

(a) GSM8K dataset

| Model | Prompt | Accuracy (%) train | Accuracy (%) test | Avg Latency (sec/query) | Avg Monetary Price ($/query) train | Avg Monetary Price ($/query) test |
|---|---|---|---|---|---|---|
| Llama-2-7b-chat | CoT few-shot | 23.36 | 23.65 | 8 | n/a | n/a |
| Llama-2-13b-chat | CoT few-shot | 37.90 | 37.91 | 16 | n/a | n/a |
| MetaMath-7b | domain expert | 92.48 | 66.19 | 8 | n/a | n/a |
| MetaMath-13b | domain expert | 92.81 | 70.43 | 16 | n/a | n/a |
| GPT-3.5-turbo (old) | domain expert | 76.60 | 73.62 | 6 | $3.83 \times 10^{-4}$ | $3.66 \times 10^{-4}$ |
| | CoT few-shot | 82.00 | 79.15 | 6 | $1.37 \times 10^{-3}$ | $1.38 \times 10^{-3}$ |
| GPT-3.5-turbo (new) | domain expert | 76.60 | 73.62 | 6 | $3.38 \times 10^{-4}$ | $3.20 \times 10^{-4}$ |
| | CoT few-shot | 82.00 | 79.15 | 6 | $9.87 \times 10^{-4}$ | $9.90 \times 10^{-4}$ |
| GPT-4-turbo | domain expert | 88.18 | 88.48 | 6 | $5.88 \times 10^{-3}$ | $5.91 \times 10^{-3}$ |
| | CoT few-shot | 92.61 | 92.34 | 6 | $1.21 \times 10^{-2}$ | $1.22 \times 10^{-2}$ |
| GPT-4 | domain expert | 84.33 | 83.17 | 6 | $7.30 \times 10^{-3}$ | $7.57 \times 10^{-3}$ |
| | CoT few-shot | 93.59 | 92.95 | 6 | $2.92 \times 10^{-2}$ | $2.94 \times 10^{-2}$ |

(b) CSQA dataset

| Model | Prompt | Accuracy (%) train | Accuracy (%) test | Avg Latency (sec/query) | Avg Monetary Price ($/query) train | Avg Monetary Price ($/query) test |
|---|---|---|---|---|---|---|
| Llama-2-7b-chat | CoT few-shot | 64.72 | 67.65 | 16 | n/a | n/a |
| Llama-2-13b-chat | CoT few-shot | 68.19 | 71.17 | 31 | n/a | n/a |
| GPT-3.5-turbo | standard few-shot | 74.09 | 76.82 | 0.3 | $6.02 \times 10^{-4}$ | $6.02 \times 10^{-4}$ |
| GPT-4 | standard few-shot | 84.29 | 87.14 | 0.7 | $1.20 \times 10^{-2}$ | $1.20 \times 10^{-2}$ |

(c) LLC dataset

| Model | Prompt | Accuracy (%) train | Accuracy (%) test | Avg Latency (sec/query) | Avg Monetary Price ($/query) train | Avg Monetary Price ($/query) test |
|---|---|---|---|---|---|---|
| Llama-2-7b-chat | CoT few-shot | 44.23% | 44.6% | 16 | n/a | n/a |
| GPT-3.5-turbo | plain text | 62.71% | 63.20% | 0.3 | $1.20 \times 10^{-4}$ | $1.14 \times 10^{-4}$ |
| GPT-3.5-turbo | CoT few-shot | 86.53% | 87.13% | 0.3 | $5.83 \times 10^{-4}$ | $5.82 \times 10^{-4}$ |
| GPT-4 | CoT few-shot | 92.68% | 93.2% | 0.7 | $1.29 \times 10^{-2}$ | $1.29 \times 10^{-2}$ |

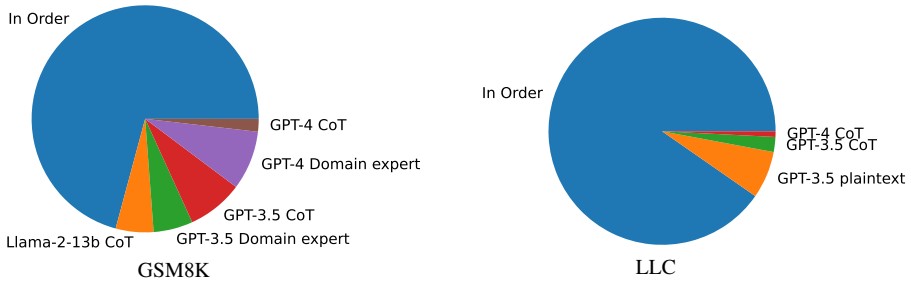

Figure 14: Fraction of questions that are solved by (LLM, prompt) combinations ordered from least to most powerful ("in order"). Minority slices are queries where less powerful combinations correctly answered.

## C.3 Different types of reasoning tasks

To visualize the differences between the three reasoning datasets, in Figure 14 we plotted the fraction of questions where the most powerful (LLM, prompt) combination in the sorted list correctly answered the question (the "in order" pie slice), versus those questions where a less powerful combination succeeded and a more powerful combination failed (all other slices of the pie). Interestingly for all datasets, there are minority cases where less powerful LLMs (the smaller pieces of the pie) can answer the question correctly. Such cases are most prevalent in the GSM8K dataset and least prevalent in LLC, possibly because the math questions of GSM8K are more difficult. Despite these dataset differences, TREACLE still chooses the right (LLM, prompt) combination to achieve higher accuracy in all datasets than the baselines.

## C.4 Different cost function parameters and datasets

Providers may adjust the per-token API prices, or the user may value latency and monetary price differently. Therefore, we conducted experiments using different settings of the $\alpha$ (defined in Appendix A.2)and $\beta$ (defined in Section 3) parameters in the cost function. In Figure 15, the cost ratio $\alpha$ increases from left to right, and hence the cost difference between more powerful (GPT) and weaker (Llama) models gradually decreases according to the definition. Under different pricing policies, TREACLE consistently achieves better performance than the online baselines. In other words, a single TREACLE model can easily accommodate varying budget requirements and cost functions, since it was trained under heterogeneous parameter settings.

Also, we mainly show GSM8K results in the main paper, because of limited space. Across the additional datasets in Figure 15 (CSQA and LLC), the results consistently show good performance.

Table 4: Overview of average cost ($) per query for different models and prompting strategies' combinations in different pricing strategies.

(a) GSM8K dataset

| Pricing Strategy | Llama-2-7b CoT | Llama-2-13b CoT | GPT-3.5-turbo (old) Domain Expert | GPT-3.5-turbo (old) CoT | GPT-4 Domain Expert | GPT-4 CoT |
|---|---|---|---|---|---|---|
| Pure monetary, $\alpha = 10$ | $1.61 \times 10^{-5}$ | $1.67 \times 10^{-4}$ | $3.66 \times 10^{-4}$ | $1.38 \times 10^{-3}$ | $7.57 \times 10^{-3}$ | $2.94 \times 10^{-2}$ |
| Pure monetary, $\alpha = 20$ | $4.01 \times 10^{-6}$ | $8.36 \times 10^{-5}$ | $3.66 \times 10^{-4}$ | $1.38 \times 10^{-3}$ | $7.57 \times 10^{-3}$ | $2.94 \times 10^{-2}$ |
| Pure monetary, $\alpha = 50$ | $6.42 \times 10^{-7}$ | $3.35 \times 10^{-5}$ | $3.66 \times 10^{-4}$ | $1.38 \times 10^{-3}$ | $7.57 \times 10^{-3}$ | $2.94 \times 10^{-2}$ |
| Price-latency combo, $\beta = 50K$ | $1.60 \times 10^{-5}$ | $3.20 \times 10^{-4}$ | $4.86 \times 10^{-4}$ | $1.50 \times 10^{-3}$ | $7.69 \times 10^{-3}$ | $2.95 \times 10^{-2}$ |
| Price-latency combo, $\beta = 500K$ | $1.60 \times 10^{-5}$ | $3.20 \times 10^{-5}$ | $3.78 \times 10^{-4}$ | $1.39 \times 10^{-3}$ | $7.58 \times 10^{-3}$ | $2.94 \times 10^{-2}$ |
| Price-latency combo, $\beta = 1000K$ | $8.00 \times 10^{-6}$ | $1.60 \times 10^{-5}$ | $3.72 \times 10^{-4}$ | $1.38 \times 10^{-3}$ | $7.58 \times 10^{-3}$ | $2.94 \times 10^{-2}$ |

| Pricing Strategy | MetaMath-7b Domain Expert | MetaMath-13b Domain Expert | GPT-3.5-turbo (new) Domain Expert | GPT-3.5-turbo (new) CoT | GPT-4-turbo Domain Expert | GPT-4-turbo CoT |
|---|---|---|---|---|---|---|
| Pure monetary, $\alpha = 10$ | $4.97 \times 10^{-6}$ | $5.01 \times 10^{-5}$ | $3.20 \times 10^{-4}$ | $9.90 \times 10^{-4}$ | $5.91 \times 10^{-3}$ | $1.22 \times 10^{-2}$ |
| Pure monetary, $\alpha = 20$ | $1.24 \times 10^{-6}$ | $2.50 \times 10^{-5}$ | $3.20 \times 10^{-4}$ | $9.90 \times 10^{-4}$ | $5.91 \times 10^{-3}$ | $1.22 \times 10^{-2}$ |
| Pure monetary, $\alpha = 50$ | $1.99 \times 10^{-7}$ | $1.00 \times 10^{-5}$ | $3.20 \times 10^{-4}$ | $9.90 \times 10^{-4}$ | $5.91 \times 10^{-3}$ | $1.22 \times 10^{-2}$ |
| Price-latency combo, $\beta = 50K$ | $1.60 \times 10^{-4}$ | $3.20 \times 10^{-4}$ | $4.40 \times 10^{-4}$ | $1.11 \times 10^{-3}$ | $6.03 \times 10^{-3}$ | $1.23 \times 10^{-2}$ |
| Price-latency combo, $\beta = 500K$ | $1.60 \times 10^{-5}$ | $3.20 \times 10^{-5}$ | $3.32 \times 10^{-4}$ | $1.00 \times 10^{-3}$ | $5.92 \times 10^{-3}$ | $1.22 \times 10^{-2}$ |
| Price-latency combo, $\beta = 1000K$ | $8.00 \times 10^{-6}$ | $1.60 \times 10^{-5}$ | $3.26 \times 10^{-4}$ | $9.96 \times 10^{-4}$ | $5.92 \times 10^{-3}$ | $1.22 \times 10^{-2}$ |

(b) CSQA dataset

| Pricing Strategy | Llama-2-7b CoT | Llama-2-13b CoT | GPT-3.5-turbo Standard | GPT-4 Standard |
|---|---|---|---|---|
| Pure monetary, $\alpha = 10$ | $1.98 \times 10^{-5}$ | $1.98 \times 10^{-4}$ | $6.02 \times 10^{-4}$ | $1.20 \times 10^{-2}$ |
| Pure monetary, $\alpha = 20$ | $4.96 \times 10^{-6}$ | $9.92 \times 10^{-5}$ | $6.02 \times 10^{-4}$ | $1.20 \times 10^{-2}$ |
| Pure monetary, $\alpha = 50$ | $7.94 \times 10^{-7}$ | $3.97 \times 10^{-5}$ | $6.02 \times 10^{-4}$ | $1.20 \times 10^{-2}$ |
| Price-latency combo, $\beta = 50K$ | $3.20 \times 10^{-4}$ | $6.20 \times 10^{-4}$ | $6.08 \times 10^{-4}$ | $1.20 \times 10^{-2}$ |
| Price-latency combo, $\beta = 500K$ | $3.20 \times 10^{-5}$ | $6.20 \times 10^{-5}$ | $6.02 \times 10^{-4}$ | $1.20 \times 10^{-2}$ |
| Price-latency combo, $\beta = 1000K$ | $1.60 \times 10^{-5}$ | $3.10 \times 10^{-5}$ | $6.02 \times 10^{-4}$ | $1.20 \times 10^{-2}$ |

(c) LLC dataset

| Pricing Strategy | Llama-2-7b CoT | GPT-3.5-turbo plaintext | GPT-3.5-turbo CoT | GPT-4 CoT |
|---|---|---|---|---|
| Pure monetary, $\alpha = 10$ | $6.60 \times 10^{-6}$ | $1.20 \times 10^{-4}$ | $5.83 \times 10^{-4}$ | $1.29 \times 10^{-2}$ |
| Pure monetary, $\alpha = 20$ | $1.65 \times 10^{-6}$ | $1.20 \times 10^{-4}$ | $5.83 \times 10^{-4}$ | $1.29 \times 10^{-2}$ |
| Pure monetary, $\alpha = 50$ | $2.64 \times 10^{-7}$ | $1.20 \times 10^{-4}$ | $5.83 \times 10^{-4}$ | $1.29 \times 10^{-2}$ |
| Price-latency combo, $\beta = 150K$ | $1.07 \times 10^{-4}$ | $1.22 \times 10^{-4}$ | $5.85 \times 10^{-4}$ | $1.29 \times 10^{-2}$ |
| Price-latency combo, $\beta = 1500K$ | $1.07 \times 10^{-5}$ | $1.20 \times 10^{-4}$ | $5.83 \times 10^{-4}$ | $1.29 \times 10^{-2}$ |
| Price-latency combo, $\beta = 3000K$ | $5.33 \times 10^{-6}$ | $1.20 \times 10^{-4}$ | $5.83 \times 10^{-4}$ | $1.29 \times 10^{-2}$ |

## C.5 Ablation experiments

We also run ablation experiments showing that prompt selection is useful, compared to using a fixed prompt (*e.g.*, CoT). The results are shown in Figure 16, where TREACLE outperforms "TREACLE (CoT only)", indicating that the ability to choose the prompt helps.

## C.6 Additional new LLM experiments

In this subsection, we report additional results relating to Section 5.2.1. The performance of the fine-tuned models with the API price adjustments or the improved open-source LLM is shown in Figure 17 ("Finetuned:GPT" and "Finetuned:Llama", respectively). The results show that the fine-tuned model with both improvements ("Finetuned:all", same as Figure 6) performs the best. The sample efficiency results for fine-tuning these models with both types of changes (corresponding to "Finetuned:all") are shown in Figure 17.

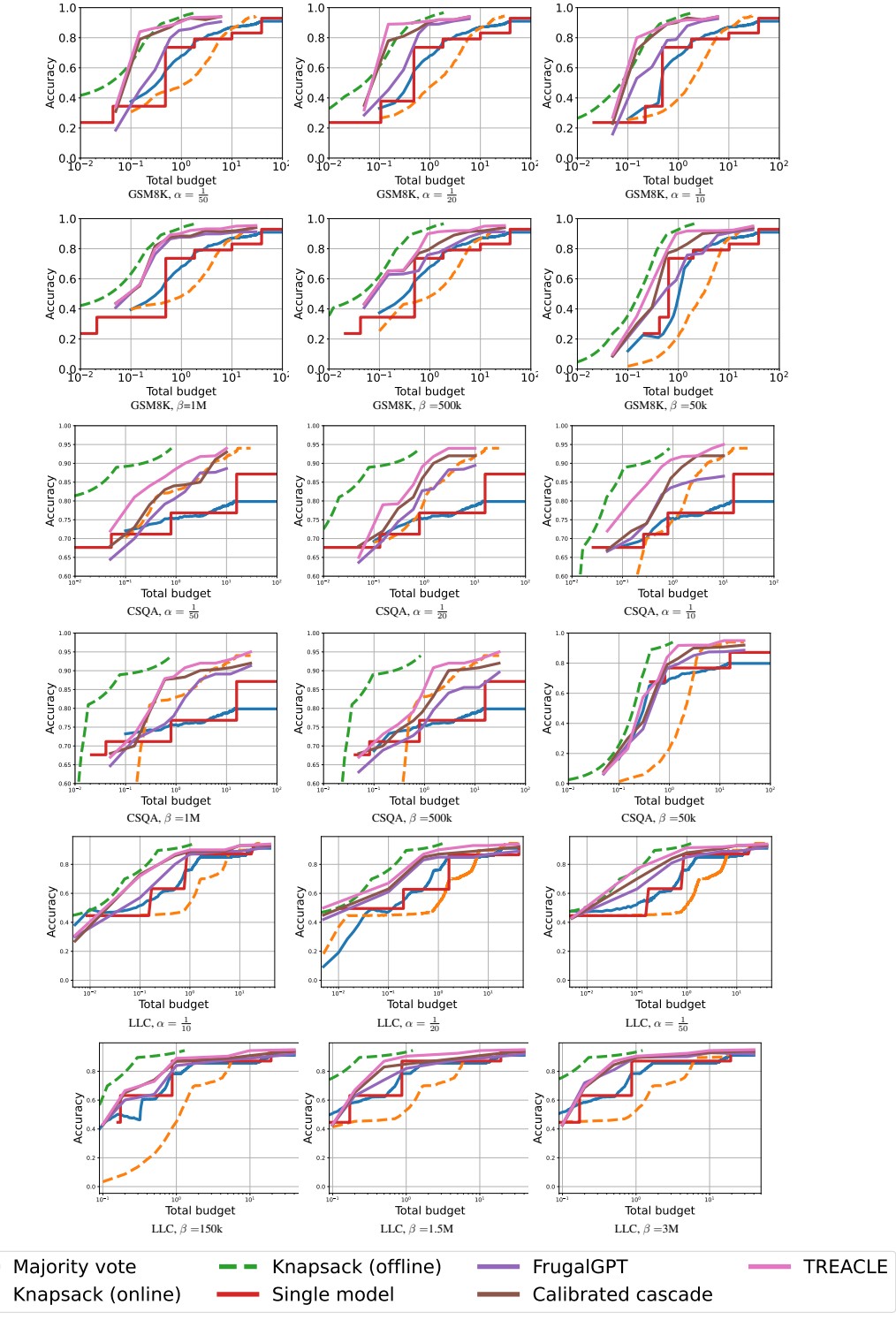

Figure 15: The performance of various methods for different cost functions and budget constraints. The dashed lines are methods that have ground knowledge, which is impractical but illustrates the best achievable performance.

Table 5: Overview of average accuracy of different models and different prompting strategies' combinations. In GSM8K table, simple CoT few-shot and complex CoT few-shot mean CoT few-shot prompts with easy and hard examples.

(a) GSM8K dataset

| Model | System Prompt | Content Prompt | Training Set Accuracy (%) | Testing Set Accuracy (%) | avg input length (Training/Testing) | avg output length (Training/Testing) |
|---|---|---|---|---|---|---|
| Llama-2-7b | "Follow example" | simple CoT few-shot | 23.36 | 23.65 | 909.81/911.43 | 120.49/119.14 |
| Llama-2-13b | NA | simple CoT few-shot | 35.65 | 33.81 | 827.81/829.43 | 218.42/214.38 |
|  | domain Eexpert | plain text | 4.47 | 25.70 | 90.15/83.43 | 28.83/130.51 |
|  | "Follow example" | simple CoT few-shot | 37.90 | 37.91 | 909.81/911.43 | 128.41/128.29 |
|  | "Follow example" | complex CoT few-shot | 42.77 | 44.05 | 2943.81/2945.43 | 328.99/326.11 |
| GPT-3.5-turbo | domain expert | plain text | 76.60 | 73.62 | 88.31/90.98 | 125.07/114.58 |
|  | "Follow example" | simple CoT few-shot | 82.00 | 79.15 | 772.05/773.70 | 107.23/108.31 |
|  | "Follow example" | complex CoT few-shot | 83.30 | 82.94 | 2419.00/2416.00 | 82.00/49.00 |
| GPT-4-turbo | domain expert | plain text | 88.18 | 88.48 | 87.31/88.98 | 166.97/167.41 |
|  | "Follow example" | simple CoT few-shot | 92.61 | 92.34 | 770.05/771.70 | 146.83/149.67 |
| GPT-4 | domain expert | plain text | 84.33 | 83.17 | 87.31/88.98 | 78.00/81.73 |
|  | "Follow example" | simple CoT few-shot | 93.95 | 92.95 | 770.05/771.70 | 101.51/103.62 |
| MetaMath-7b | domain expert | plain text | 92.48 | 66.19 | 109.15/110.80 | 48.03/54.68 |
| MetaMath-13b | domain expert | plain text | 92.81 | 70.43 | 109.15/110.80 | 47.26/56.36 |
| PaLM | "Follow example" | simple CoT few-shot | 63.05 | 62.17 | 860.04/861.70 | 115.15/115.51 |
|  | "Follow example" | complex CoT few-shot | 66.95 | 64.14 | 1918.04/1919.70 | 110.68/110.69 |

(b) CSQA dataset

| Model | System Prompting | Content Prompting | Training Set Accuracy (%) | Testing Set Accuracy (%) | avg input length (Training/Testing) | avg output length (Training/Testing) |
|---|---|---|---|---|---|---|
| Llama-2-7b | NA | plain text | 33.76 | 34.23 | 52.36/52.06 | 86.44/81.03 |
|  | "Follow example" | standard few-shot | 58.86 | 63.06 | 446.36/446.06 | 512.00/512.00 |
|  | "Follow example" | CoT few-shot | 64.72 | 67.65 | 640.36/640.06 | 512.00/512.00 |
| Llama-2-13b | NA | plain text | 29.45 | 32.10 | 52.36/52.06 | 287.60/287.34 |
|  | "Follow example" | standard few-shot | 65.29 | 66.83 | 446.36/446.06 | 512.00/511.80 |
|  | "Follow example" | CoT few-shot | 68.19 | 71.17 | 640.30/640.06 | 512.00/512.00 |
| GPT-3.5-turbo | NA | plain text | 71.37 | 73.96 | 56.74/56.44 | 4.48/4.49 |
|  | "Follow example" | standard few-shot | 74.09 | 76.82 | 396.74/396.44 | 3.50/3.55 |
|  | "Follow example" | CoT few-shot | 68.31 | 68.55 | 575.74/575.44 | 16.51/16.74 |
| GPT-4 | NA | plain text | 79.86 | 83.46 | 56.74/56.44 | 4.71/4.69 |
|  | "Follow example" | standard few-shot | 84.29 | 87.14 | 396.74/396.44 | 2.00/2.00 |
|  | "Follow example" | CoT few-shot | 82.12 | 85.83 | 575.74/575.44 | 5.05/5.20 |

(c) LLC dataset

| Model | System Prompting | Content Prompting | Training Set Accuracy (%) | Testing Set Accuracy (%) | avg input length (Training/Testing) | avg output length (Training/Testing) |
|---|---|---|---|---|---|---|
| Llama-2-7b | NA | plain text | 0.06 | 0.13 | 26.25/26.17 | 51.74/51.61 |
|  | "Follow example" | standard few-shot | 0.94 | 1.4 | 155.25/358.24 | 155.17/352.73 |
|  | "Follow example" | CoT few-shot | 44.23 | 44.60 | 344.25/344.17 | 71.77/71.11 |
| Llama-2-13b | NA | plain text | 9.01 | 9.73 | 26.25/26.17 | 55.01/54.55 |
|  | "Follow example" | standard few-shot | 2.41 | 2.93 | 155.25/155.17 | 239.28/243.78 |
|  | "Follow example" | CoT few-shot | 48.63 | 48.87 | 344.25/491.84 | 71.77/493.51 |
| GPT-3.5-turbo | NA | plain text | 62.71 | 63.20 | 30.51/30.45 | 36.93/34.08 |
|  | "Follow example" | standard few-shot | 8.16 | 9.47 | 138.51/138.45 | 5.54/5.51 |
|  | "Follow example" | CoT few-shot | 87.13 | 86.53 | 304.51/304.45 | 63.01/62.86 |
| GPT-4 | NA | plain text | 80.54 | 81.73 | 30.51/29.92 | 36.93/30.24 |
|  | "Follow example" | standard few-shot | 23.74 | 24.27 | 138.51/138.45 | 5.72/5.72 |
|  | "Follow example" | CoT few-shot | 92.68 | 93.2 | 304.51/304.45 | 63.00/62.86 |

# D  API prices

Table 4 shows further details on the parameters used in the cost functions described in Section 3. Table 6 shows the change of the GPT API's monetary price in different API versions, relating to Section 5.2.1.

Table 6: API price GPT API price of different versions, where NA means no corresponding model at that version. The price unit is $/1K tokens. In our experiment setting, old version refers to the version of 0613 and the new version refers to the version 1106.

|  | 0613 | | 1106 | | 0125 | |
|---|---|---|---|---|---|---|
|  | input | output | input | output | input | output |
| GPT-3.5-turbo | 0.0015 | 0.002 | 0.001 | 0.002 | 0.0005 | 0.002 |
| GPT-4-turbo | NA | | 0.01 | 0.03 | 0.01 | 0.03 |
| GPT-4 | 0.06 | 0.12 | 0.06 | 0.12 | 0.06 | 0.12 |

# E  Full Prompts

In this section, we show our full prompts.

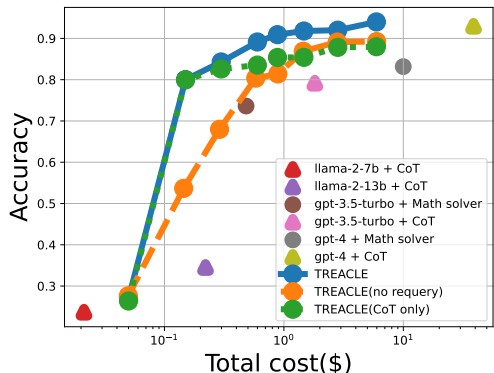

Figure 16: Ablation study. Without TREACLE's re-query and prompt selection, the performance decreases dramatically.

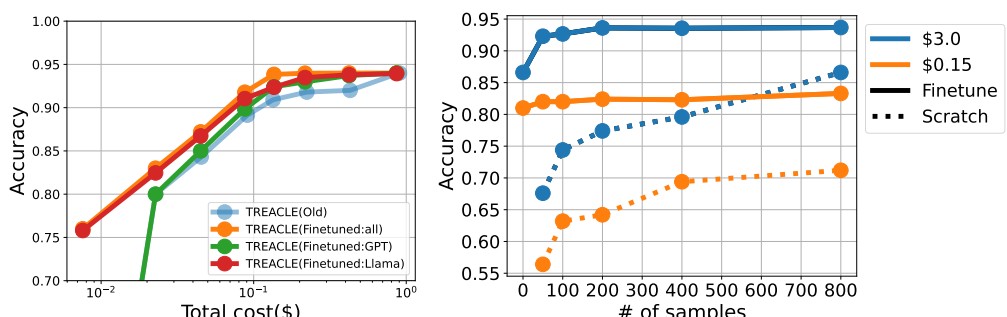

Figure 17: Additional new LLM results. Left: Zoomed in view of the accuracy with new GPT models, new Llama models, or both. Right: Sample efficiency with both new GPT models and new LLama models.

Table 7: Domain expert prompting strategy ("Math solver" and "Math assistant") in GSM8K dataset, where {Question} means that original question text.

| System Prompt | User Content Prompt |
|---|---|
| You are a math solver. Give the answer to the following question. | {Question} |
| ### Instruction:
You are a math assistant. Solve the following problem.
### Problem:
{User Content Prompt}
### Answer:
Let's think step by step. | {Question} |

Table 8: Chain-of-Thought (CoT) few-shot prompting strategy in GSM8K dataset, where {Question} means that original question text.

| System Prompt | User Content Prompt |
|---|---|
| Follow the given examples and answer the question. | Question: There are 15 trees in the grove. Grove workers will plant trees in the grove today. After they are done, there will be 21 trees. How many trees did the grove workers plant today?
Let's think step by step
There are 15 trees originally.
Then there were 21 trees after some more were planted.
So there must have been 21 - 15 = 6.
The answer is 6.

Question: If there are 3 cars in the parking lot and 2 more cars arrive, how many cars are in the parking lot?
Let's think step by step
There are originally 3 cars.
2 more cars arrive.
3 + 2 = 5.
The answer is 5.

Question: Leah had 32 chocolates and her sister had 42. If they ate 35, how many pieces do they have left in total?
Let's think step by step
Originally, Leah had 32 chocolates.
Her sister had 42.
So in total they had 32 + 42 = 74.
After eating 35, they had 74 - 35 = 39.
The answer is 39.

Question: Jason had 20 lollipops. He gave Denny some lollipops. Now Jason has 12 lollipops. How many lollipops did Jason give to Denny?
Let's think step by step
Jason started with 20 lollipops.
Then he had 12 after giving some to Denny.
So he gave Denny 20 - 12 = 8.
The answer is 8.

Question: Shawn has five toys. For Christmas, he got two toys each from his mom and dad. How many toys does he have now?
Let's think step by step
Shawn started with 5 toys.
If he got 2 toys each from his mom and dad, then that is 4 more toys.
5 + 4 = 9.
The answer is 9.

Question: There were nine computers in the server room. Five more computers were installed each day, from monday to thursday. How many computers are now in the server room?
Let's think step by step
There were originally 9 computers.
For each of 4 days, 5 more computers were added.
So 5 * 4 = 20 computers were added.
9 + 20 is 29.
The answer is 29.

Question: Michael had 58 golf balls. On tuesday, he lost 23 golf balls. On wednesday, he lost 2 more. How many golf balls did he have at the end of wednesday?
Let's think step by step
Michael started with 58 golf balls.
After losing 23 on tues- day, he had 58 - 23 = 35.
After losing 2 more, he had 35 - 2 = 33 golf balls.
The answer is 33.

Question: Olivia has $23. She bought five bagels for $3 each. How much money does she have left?
Let's think step by step
Olivia had 23 dollars.
5 bagels for 3 dollars each will be 5 x 3 = 15 dollars.
So she has 23 - 15 dollars left.
23 - 15 is 8.
The answer is 8.

Question: {Question} |

Table 9: Standard few-shot prompting strategy in CSQA dataset, where {Question} means that original question text.

| User Content Prompt |
| --- |
| Q: What do people use to absorb extra ink from a fountain pen? Answer Choices: (A) shirt pocket (B) calligrapher's hand (C) inkwell (D) desk drawer (E) blotter
A: The answer is E.

Q: What home entertainment equipment requires cable? Answer Choices: (A) radio shack (B) substation (C) television (D) cabinet
A: The answer is C.

Q: The fox walked from the city into the forest, what was it looking for? Answer Choices: (A) pretty flowers (B) hen house (C) natural habitat (D) storybook
A: The answer is B.

Q: Sammy wanted to go to where the people were. Where might he go? Answer Choices: (A) populated areas (B) race track (C) desert (D) apartment (E) roadblock
A: The answer is A.

Q: Where do you put your grapes just before checking out? Answer Choices: (A) mouth (B) grocery cart (C)supermarket (D) fruit basket (E) fruit market
A: The answer is B.

Q: Google Maps and other highway and street GPS services have replaced what? Answer Choices: (A) united states (B) mexico (C) countryside (D) atlas
A: The answer is D.

Q: Before getting a divorce, what did the wife feel who was doing all the work? Answer Choices: (A) harder (B) anguish (C) bitterness (D) tears (E) sadness
A: The answer is C.

Q: {Question }
A: The answer is |

Table 10: Chain-of-Thought (CoT) few-shot prompting strategy in CSQA dataset, where {Question} means that original question text.

| User Content Prompt |
| --- |
| Q: What do people use to absorb extra ink from a fountain pen? Answer Choices: (A) shirt pocket (B) calligrapher's hand (C) inkwell (D) desk drawer (E) blotter
A: The answer must be an item that can absorb ink. Of the above choices, only blotters are used to absorb ink. The answer is E.

Q: What home entertainment equipment requires cable? Answer Choices: (A) radio shack (B) substation (C) television (D) cabinet
A: The answer must require cable. Of the above choices, only television requires cable. The answer is C.

Q: The fox walked from the city into the forest, what was it looking for? Answer Choices: (A) pretty flowers (B) hen house (C) natural habitat (D) storybook
A: The answer must be something in the forest. Of the above choices, only natural habitat is in the forest. The answer is B.

Q: Sammy wanted to go to where the people were. Where might he go? Answer Choices: (A) populated areas (B) race track (C) desert (D) apartment (E) roadblock
A: The answer must be a place with a lot of people. Of the above choices, only populated areas have a lot of people. The answer is A.

Q: Where do you put your grapes just before checking out? Answer Choices: (A) mouth (B) grocery cart (C)supermarket (D) fruit basket (E) fruit market
A: The answer should be the place where grocery items are placed before checking out. Of the above choices, grocery cart makes the most sense for holding grocery items. The answer is B.

Q: Google Maps and other highway and street GPS services have replaced what? Answer Choices: (A) united states (B) mexico (C) countryside (D) atlas
A: The answer must be something that used to do what Google Maps and GPS services do, which is to give directions. Of the above choices, only atlases are used to give directions. The answer is D.

Q: Before getting a divorce, what did the wife feel who was doing all the work? Answer Choices: (A) harder (B) anguish (C) bitterness (D) tears (E) sadness
A: The answer should be the feeling of someone getting divorced who was doing all the work. Of the above choices, the closest feeling is bitterness. The answer is C.

Q: {Question }
A: |

Table 11: Standard few-shot prompting strategy in LLC dataset, where {Question} means that original question text.

| User Content Prompt |
| --- |
| Question: Take the last letters of the words in "Elon Musk" and concatenate them. The answer is nk. |
| Question: Take the last letters of the words in "Larry Page" and concatenate them. The answer is ye. |
| Question: Take the last letters of the words in "Sergey Brin" and concatenate them. The answer is yn. |
| Question: Take the last letters of the words in "Bill Gates" and concatenate them. The answer is ls. |
| Question: {Question} |

Table 12: Chain-of-Thought (CoT) few-shot prompting strategy in LLC dataset, where {Question} means that original question text.

| User Content Prompt |
| --- |
| Question: Take the last letters of the words in "Elon Musk" and concatenate them. Let's think step by step. The last letter of "Elon" is "n". The last letter of "Musk" is "k". Concatenating them is "nk". The answer is nk. |
| Question: Take the last letters of the words in "Larry Page" and concatenate them. Let's think step by step. The last letter of "Larry" is "y". The last letter of "Page" is "e". Concatenating them is "ye". The answer is ye. |
| Question: Take the last letters of the words in "Sergey Brin" and concatenate them. Let's think step by step. The last letter of "Sergey" is "y". The last letter of "Brin" is "n". Concatenating them is "yn". The answer is yn. |
| Question: Take the last letters of the words in "Bill Gates" and concatenate them. Let's think step by step. The last letter of "Bill" is "l". The last letter of "Gates" is "s". Concatenating them is "ls". The answer is ls. |
| Question: {Question} Let's think step by step. |

