# OpenReview forum: "Efficient Contextual LLM Cascades through Budget-Constrained Policy Learning"
_NeurIPS.cc/2024/Conference — NeurIPS 2024 poster_

### Official Review · Reviewer_gLy5 · 2024-06-15

**Soundness:** 3
**Presentation:** 3
**Contribution:** 2
**Rating:** 5
**Confidence:** 2

**Summary:**

The paper introduces TREACLE, a reinforcement learning policy designed to optimize the selection of LLMs and prompting schemes based on a user's budget constraints in terms of cost and latency.

**Strengths:**

1. TREACLE enables substantial cost savings compared to existing methods by intelligently choosing among various LLMs and prompts based on the monetary cost, latency, and accuracy.

2. By considering the context of the question, including embeddings and past interactions, TREACLE customizes the prompting strategy to balance accuracy against cost, often using advanced prompting strategies like Chain-of-Thought to improve answer quality at a controlled cost.

3. The system dynamically decides whether to re-query a model based on the current response's consistency and the remaining budget, which helps in refining the answers further without exceeding budget constraints.

**Weaknesses:**

1. Dynamic selection of models and re-querying could lead to increased computational costs and delays, especially in scenarios requiring high real-time performance. Although the system is designed to save costs, frequent model switching and complex queries might backfire.

2. The reward mechanism mentioned in the text depends on accurate answer feedback to adjust strategies, but in practical applications, users may not always provide clear or consistent feedback. This could lead to instability during the learning process and inaccuracies in reward distribution.

3. My biggest concern is that the architecture of the LLM itself has not been changed. It merely adds additional reinforcement learning, which seems overly reliant on data, and might perform poorly on new types of questions or unseen data, limiting the model's generalizability.

**Questions:**

1. Is the setting in lines 103-107 somewhat far from reality? The same question is asked many times; wouldn't that mean, in practice, there is an endless number of questions that need to be learned, and indeed, the same question could be asked in many different ways. Also, users might not always provide feedback on a question, so how would one obtain the reward?
2. How large is the actual action space, particularly with the three different prompt strategies (standard, domain expert, and CoT)?
3. Optimizing the selection of LLMs and prompting schemes with constraints is an intuitive idea. There are already some similar works:
(1)- "Which LLM to Play? Convergence-Aware Online Model Selection with Time-Increasing Bandits"
(2)- "Cost-Effective Online Multi-LLM Selection with Versatile Reward Models"
(3)- "Best Arm Identification for Prompt Learning under a Limited Budget"
So, the Table 1 listed by the authors is not comprehensive. If the authors do not have time to conduct new experiments for comparison, you could provide a textual description comparing these works, highlighting the advantages of your own work.
4. Where are "finetune" and "scratch" mentioned in Figure 7?

**Limitations:**

As mentioned by the authors, the RL policy’s budget does not account for the cost of collecting the training data.

---

> ### Author Rebuttal · Authors · 2024-08-07
>
> ### W1: Dynamic selection of models and re-querying could lead to increased computational costs and delays
>
> As the reviewer points out, latency plays a crucial role in real-time settings (e.g., voice assistants). To address scenarios requiring real-time performance, we incorporated latency as a component of the cost constraint (see L142). TREACLE allows users to choose the trade-off between monetary cost and latency by adjusting a trade-off coefficient $\beta$, where $cost=\text{latency} + \beta * \text{monetary price}$. This enables the users to choose the balance between monetary cost and latency according to their specific needs. The rightmost two subfigures in Figure 3 show the performance when latency is included in  the cost constraint, using real end-to-end latency values (including computation and communication) that we measured by querying Llama and GPT models (see Figure 13 in the Appendix).
> Further experiments regarding latency are described in Appendices C.1 and C.4. We conducted experiments when the API query latency varied over time; these results are summarized in Table 2 in Appendix C.1 and show that TREACLE can adapt to API querying latency that differs across models and time. Figure 15 in Appendix C.4 shows results with latency in the cost function for more datasets and $\beta$ values.
> Finally, an additional component of computation latency is the RL model itself. Our RL model is a two-layer neural network, so the inference time is very fast (on the order of ms) and hence negligible.
>
> ### W2: Users may not always provide clear or consistent feedback.
>
> To clarify, we do not rely on user feedback. Rather, during the training phase, the RL learns a policy to maximize rewards using the correct answers from the datasets, and executes that trained policy during the test phase (see L197). The main form of “feedback” during the test phase is the consistency of the responses returned by the LLMs, as in prior work [20]. However, we found that response consistency alone is insufficient feedback, so TREACLE combines response consistency with prompt text embedding, for the first time, to enhance the overall effectiveness. Adding user feedback, along the lines of active learning, could be an interesting future extension for our framework.
>
> ### W3: The architecture of the LLM itself has not been changed
>
> We believe that adding reinforcement learning on top of existing LLMs is a strength of our framework, since the modular design enables incorporating new LLMs that are constantly emerging. TREACLE’s framework is generalizable and we investigated its adaptability to new types of questions (Section 5.2.3), unseen data with harder questions (Section 5.2.2), and new LLMs (Section 5.2.1).
> For example, to understand whether TREACLE can adapt to new types of questions, we conducted experiments (L384-394). The base model is trained using commonsense reasoning questions (CSQA dataset), and the new unseen question type are math problems (GSM8K dataset). The results in Figure 10b show that TREACLE can achieve high accuracy on new question types with only a minimal amount of extra training (i.e., “Fine-tune on 200 GSM8K” is close to “Train on GSM8K”). This minimal amount of extra training is done by freezing the base RL policy trained on CSQA and fine-tuning the “text embedding” feature in the state vector using GSM8K.
>
> ### Q1: Is the setting in lines 103-107 somewhat far from reality?
>
> The setting of re-querying the same question multiple times has been established in the literature [16,20]. In practice, there are a limited number of unique combinations of language models and prompts. In our experiments, there are 6 possible combinations. The reinforcement learning (RL) model is designed to be general; it does not need to be trained with every possible question variation and answer to choose the best action.
>
> ### Q2: How large is the actual action space?
>
> There are 3 possible actions in the action space, no matter which prompt or model was used on the previous query: Return the current response, re-query with the same model-prompt pair, or select a new model-prompt pair (from the next option in the cascade, not choosing from all possible model-prompt combinations). We will elaborate on this in L188.
> ### Q3: Related work
>
> Compared to our work, firstly, we are the only work that considers latency constraints. Some scenarios require high real-time performance, as mentioned by Reviewer gLy5. Second, we take into account both LLMs and prompts, since performance and cost are influenced by both factors. The related papers only focus on one of them. Third, we show generalization to unseen tasks and that our method is pretty sample efficient. This is related to the online decision exploration cost mentioned in (2).  Fourth, unlike the mentioned papers, we treat the LLMs and prompt pair as a black box, without training or fine-tuning them. Finally, we determine different model and prompt pairs for each sample, rather than for the entire task. We will add these papers to the table in the related work section, and hope to add them as baselines once their code is released.
>
> ### Q4: Where are "finetune" and "scratch" mentioned in Figure 7?
>
> “finetune” (solid blue/orange lines) means that we start with a model that was trained using the old API prices and language models (LLMs), and then fine-tune it with new state-action trajectories collected using the new API prices and LLMs. In contrast, “scratch” (dashed blue/orange lines) means that we train from scratch, i.e., we initialize the reinforcement learning (RL) model randomly and train it directly with the newly collected trajectories. We will add these clarifications to the text of the paper.
>
> ### L1: Cost of collecting training data
>
> As mentioned in the paper, we plan to release the training datasets and code for reproducibility so that others can avoid the cost of collecting training data and adapt the framework to other LLMs and tasks.

---

> > ### Comment · Reviewer_gLy5 · 2024-08-12
> >
> > Thank you for your response, which has addressed my concerns. However, as Reviewer LPQa mentioned, the main weaknesses of the paper are that the method overlooks the actual cost, and the supplementary experiment TREACLE+penalty does not show a significant effect. Overall, I choose to increase the score by one point, but my confidence is not high.

---

> ### Author Response · Authors · 2024-08-12
>
> We thank the reviewer for reassessing our work and raising their rating.
>
> Regarding Reviewer LPQa's concern: We realized that Figure 1 may not reflect the full picture due to the log-scale of the x-axis. Kindly consider the equivalent table provided below which demonstrates that TREACLE with penalty facilitates significant cost reduction.
>
> | Budget   | Total Cost (No Penalty) | Accuracy (No Penalty) | Total Cost (With Penalty) | Accuracy (With Penalty) |
> |----------|---------------------------|-------------------|--------------------|----------------------------|
> | 0.05 | 0.050                     | 0.273             | 0.047             | 0.268                       |
> | 0.15 | 0.150                     | 0.799             | 0.149              | 0.791                       |
> | 0.3   | 0.298                     | 0.843             | 0.297              | 0.842                       |
> | 0.6   | 0.599                     | 0.891             | 0.596              | 0.890                       |
> | 0.9   | 0.884                     | 0.909             | 0.876              | 0.907                       |
> | 1.5   | 1.471                     | 0.918             | 1.261              | 0.911                       |
> | 3      | 2.826                     | 0.920             | 2.471              | 0.917                       |
> | 6      | 5.951                     | 0.938             | 4.517              | 0.926                       |
>
> Above, the last data point (budget of $6) shows a **24.09% reduction in cost** compared to the original (from 5.951 to 4.517), with a corresponding 1.279% decrease in accuracy (from 93.8% to 92.6%). At a budget of $3, the improvement is also evident with **12.56%** lower costs and accuracy drop of only 0.3%. Finally, this table also demonstrates that TREACLE works as intended:
>
> - Without penalty, the method fully utilizes the available budget i.e. the actual cost is very close to the max available budget. This is in line with the goal of maximizing accuracy subject to the budget constraint.
>
> - With penalty, it utilizes the budget more efficiently (e.g. 24% cheaper) at the cost of slightly reduced accuracy. Note that such an accuracy-cost tradeoff is fundamental and not really avoidable (there is no free lunch; we cannot perfectly know which are the easy questions to save on). Also note that by increasing the penalty parameter that trades off between cost and accuracy, one can achieve a more drastic cost reduction at the expense of (slightly) worse accuracy. We are happy to provide more results with different penalty parameter $\lambda$ choices.
>
> In summary, we hope this clarifies that if we wish to minimize actual costs as pointed out by the reviewer, the model with penalty can efficiently allocate resources to avoid additional spending, when accuracy gains start to diminish (using tradeoff parameter $\lambda$). We thank the reviewer again for raising the actual spent cost as an important consideration.

---

### Official Review · Reviewer_LPQa · 2024-07-13

**Soundness:** 3
**Presentation:** 3
**Contribution:** 2
**Rating:** 4
**Confidence:** 5

**Summary:**

The paper proposes a reinforcement learning method to select the model and prompting. It combines with monetary cost and latency constraints. The design of the features contains question text embeddings and response history. Experiments studies the cost savings.

**Strengths:**

1, Important problems.
2, Interesting algorithm design
3, Many experiments

**Weaknesses:**

The main weaknesses of the paper are that the method overlooks the actual cost and the experiments are not convincing enough.

1, Insufficient to differentiate simple and difficult questions in model selection. A good model selection mechanism should choose low-cost models for simple questions and more powerful models to improve accuracy for difficult questions. The selected models should differ between simple and complex questions. From Figure 4a, it can be seen that when the budget is low (0.05), the selected models are limited to llama2-7b and llama2-13b, failing to call powerful models for solving difficult problems. However, when the budget is high (10), the selected models no longer include llama2-7b, thus missing the opportunity to use low-cost models to reduce expenditure.

2, Lack minimizing actual cost in method design. The method proposed in this paper is designed to only consider staying within the long-term (maximum) budget, without optimizing for minimal actual cost, which can lead to cost inefficiencies. This may derive from overlooking individual cost optimization.

3, Unnecessary high actual costs on simple questions. In simple questions, the method proposed in this paper shows similar accuracy to existing methods, but it consumes an additional 19% of the actual cost. Figure 8b shows that when the total (maximum) budget is \$1, the method proposed in this paper (TREACLE) performs similarly to Single model and Calibrated cascade in terms of accuracy. However, according to Figure 8c, Single model/Calibrated cascade actually costs 0.69/0.702, while TREACLE costs 0.84 (an additional 19%).

**Questions:**

Why do the authors summarize in Table 1 that MoT, a consistency-based method, cannot be robust to new models? I believe this statement is incorrect because this method is training-free. If it is due to limitations in various model capabilities, I suggest introducing the option of 'partially limited'.

**Limitations:**

It is recommended to apply the method proposed in this paper to more tasks. The tasks in this paper are too limited.

---

> ### Author Rebuttal · Authors · 2024-08-07
>
> We greatly appreciate the reviewer’s detailed and constructive feedback.
>
>
> ## Weakness 2: Main weaknesses of the paper are that the method overlooks the actual cost.
> We acknowledge the reviewer's concern and agree that our method uses a total budget constraint which minimizes the individual query costs in an indirect fashion. In general, the choice of cost function (e.g. the total budget constraint or individual costs or both) is a design decision. The constraint budget formulation we adopt is widely utilized in resource allocation problems and is typically more difficult to solve/enforce compared to unconstrained RL. That said, our approach is flexible and can also directly minimize the actual cost through a penalized reward formulation.
> Concretely, following the reviewer’s suggestion, we examined a penalized variation of TREACLE where the reward formulation is (see PDF in rebuttal for all reviewers for precise form)
> $$\mathbb{E}[query\\_acc - \lambda\cdot query\\_cost]~~~subject~to~~~total\\_cost \leq B$$
> Here, $\lambda\cdot query\\_cost$ is the penalty term we incorporate. This penalized form brings us to Weakness 3 discussed below.
>
> ## Weakness 3: Unnecessary high actual costs on simple questions.
>  We have conducted experiments on TREACLE+penalty which are provided in Figures 1 and 2 of the PDF.
>
>
> - Over the set of easy questions, penalty promotes the use of cheaper models and matches the single model baseline without loss of accuracy. This highlights that penalty aids the efficient use of budget in line with the reviewer’s intuition. The TREACLE+penalty model tends to solve easy questions with less powerful LLMs and the budget spent decreases to 0.717 which is almost the same as the single model baseline while also achieving on-par accuracy, 0.987 (TREACLE+penalty) v.s. 0.986 (single model).
>
> - In general, penalty term will result in a budget-accuracy tradeoff. This is because we don’t exactly know the optimal model for solving a particular question. When evaluating TREACLE+penalty over all queries, we find that penalty improves the budget utilization of TREACLE (actual cost decrease from 0.97 to 0.94) however it also results in (minor) accuracy degradation (accuracy decrease from 0.92 to 0.90).
>
> ## Weakness 1: Insufficient to differentiate simple and difficult questions in model selection.
>
> The results actually show that our method makes intelligent choices in a budget-aware fashion. Concretely:
>
> - When the budget is low (0.05), the model correctly prioritizes easy questions, as calling an expensive model for one difficult question would leave more than 20 other questions unanswered, harming the overall performance. For instance, with the settings of Figure 4a, if we pick 5 difficult questions and solve them with GPT-4 (CoT), and use the remaining budget for the rest, the total accuracy drops from 0.31 to 0.20.
>
> - When the budget is sufficiently high, the model opts for using powerful models for all questions to maximize acuracy. This is because relying on smaller models or difficulty estimation stage can lead to errors. For example, the average accuracy of Llama-2-7b (CoT) is only 23.65%. Forcing the model to use Llama-2-7b (CoT) decreases the performance from 92.47% to 92.20%, and for the Majority Voting baseline, from 83.62% to 83.30%. As discussed under **Weakness 2 and 3**, when we use TREACLE-penalty, the model starts prioritizing Llama-2 even with a high budget.
>
> ## Question 1: Why MoT cannot be robust to new models
> We agree that MoT is training-free; however, the original MoT paper does not address the inclusion of new models. We believe it has two limitations when incorporating new models: Firstly, MoT only allows for two models, a weak and a strong one, and there is no clear way of adding more models. Secondly, MoT uses a threshold to decide whether an answer is accepted or not. There is no guarantee that a fixed threshold will work across multiple distinct models or how to find/optimize such thresholds efficiently. For instance, our *calibrated cascade* approach is reminiscent of MoT and, empirically, we find that it is not robust to distribution change.
>
> ## Limitations: It is recommended to apply the method proposed in this paper to more tasks.
> Thank you for your suggestion. We plan to include additional tasks in future work. Currently, we have evaluated our approach using three representative datasets. Additionally, we demonstrate that our model can be seamlessly adapted to unseen tasks (L384-395, Fig 10b) and that a single model can effectively handle multiple types of tasks while maintaining a shared budget constraint (L374-383, Fig 10a).
>
>
> We genuinely appreciate the reviewer’s excellent points which motivated us to study penalized TREACLE. We will incorporate these discussions and evaluations and revise the manuscript accordingly. We hope that this response has addressed their concerns. We would be happy to engage further during the discussion week.

---

> > ### Comment · Reviewer_LPQa · 2024-08-11
> >
> > Thank you for the detailed response. I appreciate the design regarding the penalty. However, based on the newly added experiments, Figure 1 in the attachment shows that the cost has not been significantly reduced, and the accuracy has decreased. Therefore, I will maintain the original score.

---

> > > ### Author Response · Authors · 2024-08-12
> > >
> > > Thank you for your time and feedback. We realized that Figure 1 may not reflect the full picture due to the log-scale of the x-axis. Kindly consider the equivalent table provided below which demonstrates that TREACLE with penalty facilitates significant cost reduction.
> > >
> > > | Budget   | Total Cost (No Penalty) | Accuracy (No Penalty) | Total Cost (With Penalty) | Accuracy (With Penalty) |
> > > |----------|---------------------------|-------------------|--------------------|----------------------------|
> > > | 0.05 | 0.050                     | 0.273             | 0.047             | 0.268                       |
> > > | 0.15 | 0.150                     | 0.799             | 0.149              | 0.791                       |
> > > | 0.3   | 0.298                     | 0.843             | 0.297              | 0.842                       |
> > > | 0.6   | 0.599                     | 0.891             | 0.596              | 0.890                       |
> > > | 0.9   | 0.884                     | 0.909             | 0.876              | 0.907                       |
> > > | 1.5   | 1.471                     | 0.918             | 1.261              | 0.911                       |
> > > | 3      | 2.826                     | 0.920             | 2.471              | 0.917                       |
> > > | 6      | 5.951                     | 0.938             | 4.517              | 0.926                       |
> > >
> > > Above, the last data point (budget of $6) shows a **24.09% reduction in cost** compared to the original (from 5.951 to 4.517), with a corresponding 1.279% decrease in accuracy (from 93.8% to 92.6%). At a budget of $3, the improvement is also evident with **12.56%** lower costs and accuracy drop of only 0.3%. Finally, this table also demonstrates that TREACLE works as intended:
> > >
> > > - Without penalty, the method fully utilizes the available budget i.e. the actual cost is very close to the max available budget. This is in line with the goal of maximizing accuracy subject to the budget constraint.
> > >
> > > - With penalty, it utilizes the budget more efficiently (e.g. 24% cheaper) at the cost of slightly reduced accuracy. Note that such an accuracy-cost tradeoff is fundamental and not really avoidable (there is no free lunch; we cannot perfectly know which are the easy questions to save on). Also note that by increasing the penalty parameter that trades off between cost and accuracy, one can achieve a more drastic cost reduction at the expense of (slightly) worse accuracy. We are happy to provide more results with different penalty parameter $\lambda$ choices.
> > >
> > > In summary, we hope this clarifies that if we wish to minimize actual costs as pointed out by the reviewer, the model with penalty can efficiently allocate resources to avoid additional spending, when accuracy gains start to diminish (using tradeoff parameter $\lambda$). We thank the reviewer again for raising the actual spent cost as an important consideration.

---

### Official Review · Reviewer_EMUg · 2024-07-13

**Soundness:** 3
**Presentation:** 2
**Contribution:** 3
**Rating:** 5
**Confidence:** 2

**Summary:**

This paper presents a framework for managing different budgets—such as accuracy, cost, and latency—when utilizing Large Language Models (LLMs) for reasoning tasks. Recognizing that reasoning tasks can be broken down into a series of question-and-answer interactions, the authors propose a method to allocate models of varying sizes to handle these multi-round interactions effectively. To achieve this, they use a reinforcement learning approach to train a model that can estimate the budget space. The efficacy of the proposed method is validated through evaluations on standard benchmarks, including GSM8k, CSQA, and LLC.

**Strengths:**

* Introduces a novel framework for modeling multi-turn reasoning sequences, taking into account the holistic aspects of LLM cost, latency, and other factors.
* Empirically evaluates the reinforcement learning-based training and execution of the algorithm in realistic settings using popular datasets.
* Provides intriguing real-world observations (Section 5.2.1) regarding the impact of pricing changes and the introduction of new models.

**Weaknesses:**

* The current framework is inadequate if a capable model can plan ahead by considering multiple questions or a trajectory of questions in advance, even while using various models for the answers.
* The state vector is limited, as text embeddings alone may not fully capture the complete characteristics of the prompt.
* There is a need for a reliable method to estimate the quality of questions and answers generated by the model.

**Questions:**

* Many figures require a color printer and are difficult to read due to solid and translucent lines. The authors should consider improving the visuals to be more readable across different printing methods.
* Please also refer to the weaknesses section for additional feedback.

**Limitations:**

Ok.

---

> ### Author Rebuttal · Authors · 2024-08-07
>
> ##  W1: The current framework is inadequate if a capable model can plan ahead by considering multiple questions or a trajectory of questions in advance, even while using various models for the answers.
>
> An approach of considering a batch of questions at once could certainly work, whereas in our framework we consider questions one-by-one. We do this mainly for scalability, as considering multiple questions into the future would greatly increase the dimension of the state and action space and hence the amount of training needed. We believe that our framework can extend to considering batches of questions in the future and it would be interesting to explore.
>
> ##  W2: The state vector is limited, as text embeddings alone may not fully capture the complete characteristics of the prompt.
>
> We agree that text embeddings alone may not capture all characteristics of the prompt. Therefore, our state vector includes both *text embeddings* and *response consistency*, which is an indirect measure of another prompt characteristic, its difficulty (for example, difficult prompts tend to have less consistent answers [20]). We are the first to implement this combination of state features, resulting in notable performance improvements. Also, our framework is quite flexible, so additional features relating to prompt characteristics can easily be added.
>
> In greater detail, including both text embeddings and response consistency in the state vector improves performance. One baseline method, FrugalGPT, only uses text embeddings. Another baseline method, Majority Voting, only uses response consistency. We have demonstrated significant improvements of TREACLE over both baselines. We also provide theoretical justification supporting why policies considering response consistency perform well (L216-239).
>
> ##  W3: There is a need for a reliable method to estimate the quality of questions and answers generated by the model.
> Indeed, we initially had the same thought, and hence we developed the Calibrated Cascade baseline, which *explicitly* estimates answer quality (using the same state vector as TREACLE as input). It then uses the estimated answer quality to decide whether to query another LLM. While this baseline generally outperforms the other baselines, it is not as robust as TREACLE particularly when there are shifts in question difficulty, because it does not carefully consider the remaining budget when answering easy vs hard questions. In contrast, TREACLE *implicitly* estimates answer quality by combining previous answers with text embeddings of the question, in order to decide which LLMs to query.
>
>
> ##  Q1:  improving the visuals to be more readable across different printing methods
> We will improve the figures to enhance readability for different printing methods, ensuring they are clear even in black-and-white print.

---

> > ### Comment · Reviewer_EMUg · 2024-08-13
> >
> > Thank you for your rebuttal. However, I still have some concerns that remain unaddressed. The responses provided were not entirely satisfactory.
> >
> > I also share the concerns raised by other reviewers (LPQa, gLy5) regarding the absence of direct cost minimization in the modeling process. The two lines in Figure 1 of the rebuttal PDF (with and without the cost penalty) looks nearly identical. It suggests that the cost penalty term is not working effectively.
> >
> > Even in the table the authors provided in the comment, the cost and accuracy trade-off is not great. e.g. compare a) budget of 1.5 gives the original method 0.918 accuracy with cost of 1.471. b) budget of 3 with the cost penalty method gives 0.917 accuracy with the cost of 2.471.
> >
> > Given these remaining concerns, I will be adjusting my score accordingly.

---

> > > ### Author Response · Authors · 2024-08-13
> > >
> > > Thank you for your valuable feedback. With the cost-penalized formulation, tuning $\lambda$ can further adjust the cost-accuracy tradeoff. For example, with regards to the reviewer's example, for a different parameter setting ($\lambda=100, \text{budget}= 3$; previous figures were for fixed $\lambda=10$), the accuracy is 0.918 and the cost is 1.454. This is comparable to TREACLE without penalty and \$1.5 budget, which has the same accuracy of 0.918 and cost 1.471. In other words, with a larger budget and by adding the cost penalty, we can more finely control the actual cost.
> > >
> > > Overall, many papers use the budget-constrained setting [1,2,3,4]. We explored the cost penalty based on reviewer feedback and found that the TREACLE framework flexibly extended to such a formulation. The results suggest that the cost penalty formulation can perform well with some parameter tuning, which is a general disadvantage of cost-penalty formulations. The advantage of our original formulation is that it does not require parameter tuning, only a simple total budget setting, which easier for practitioners. We would like to add the cost-penalty results as a subsection to the paper further exploring this alternative formulation and different settings of $\lambda$.
> > >
> > > [1] Bai, Fan, Alan Ritter, and Wei Xu. "Pre-train or Annotate? Domain Adaptation with a Constrained Budget." Proceedings of the 2021 Conference on Empirical Methods in Natural Language Processing. 2021.
> > >
> > > [2] Hoffmann, Jordan, et al. "Training compute-optimal large language models." Proceedings of the 36th International Conference on Neural Information Processing Systems. 2022.
> > >
> > > [3] Chen, Lingjiao, Matei Zaharia, and James Zou. "Frugalgpt: How to use large language models while reducing cost and improving performance." arXiv preprint arXiv:2305.05176 (2023).
> > >
> > > [4] Shi, Chengshuai, et al. "Best arm identification for prompt learning under a limited budget." arXiv preprint arXiv:2402.09723 (2024).

---

### Official Review · Reviewer_kukn · 2024-07-16

**Soundness:** 3
**Presentation:** 3
**Contribution:** 3
**Rating:** 7
**Confidence:** 4

**Summary:**

This paper aims to solve the problem that LLMs can be costly, in particular using technologies such as COT. It proposes to apply RL to select the model and prompting scheme. Experimental results show that the proposed method can maintain the model performance while saving up to 85% costs.

**Strengths:**

1. This paper is generally well-written and is well motivated.

2. The experiments are solid with great cost savings.

3. The idea of using RL to reason over question and response embeddings is interesting.

**Weaknesses:**

No great weakness.

**Questions:**

NA

---

> ### Author Rebuttal · Authors · 2024-08-07
>
> Thank you for your positive feedback on the manuscript. We appreciate your recognition of the reinforcement-learning based approach to achieve significant cost savings when querying LLMs.

---

### Author Rebuttal · Authors · 2024-08-07

Thank you to the reviewers for their thoughtful reviews and constructive comments. We have provided individual responses to each of the reviewers. In addition to these responses, we have conducted additional experiments to evaluate cost-accuracy tradeoffs by including the cost constraint in the objective, further reducing cost (details are in the response to reviewer LPQa). We hope these new experiments and clarifications will be acceptable to the reviewers. We thank the reviewers for their valuable time.

---

### Author Response · Authors · 2024-08-13

We sincerely thank all reviewers for their detailed and constructive feedback. Since there is a shared concern, we would like to clarify that **the two curves in Figure 1 being close is the expected and the desired behavior**. The constrained and penalized optimization are dual forms, both aiming to allocate resources optimally to maximize accuracy. Thus, for the same level of actual cost, we expect that constrained and penalized forms achieve (approximately) the same accuracy. This is why the **actual-cost vs accuracy pareto-front in Figure 1 are essentially the same for the penalized and constrained forms**. This also means that **our base method can minimize the actual cost by adjusting the budget to match the desired cost level**.

In our initial response to Reviewer LPQa, we viewed the total budget as a fixed user-specified quantity. This was to highlight the fact that, by adding penalty, one can improve the utilization of this fixed budget (rather than maxing it out to increase accuracy). However, following the reviewer discussion and potential confusion, we realize that the correct viewpoint is that by adjusting the budget, the algorithm indeed minimizes the actual cost.

We hope this clarifies our contribution and Figure 1.

Authors

---

### Decision · Program_Chairs · 2024-09-25

**Decision:**

Accept (poster)

**Comment:**

This paper aims to address a practical challenge of having a set of LLMs and possible prompts with different costs, and the goal is to maximize the accuracy of an order set of questions without exceeding a given budget. The authors formalize this as an MDP with an action space by allowing an agent to repeatedly query the same model+prompt, to move to the next model+prompt option (according to a predefined accuracy/cost ordering that is pre-estimated), or to move to the next question (meaning that the last returned answer is used, or that the question is skipped if this action was the first to be taken for the question). The state space includes question text embedding, the normalized remaining budget and few other features.

Reviewers found the paper to be overall well written and interesting and appreciate the problem formulation [LPQa, EMUg, kukn], the extensive experiments [LPQa, EMUg, kukn] and allowing the agent to re-query to estimate confidence in the answer [gLy5, EMUg]. On the other hand, reviewers expressed concerns about the reward function initially used by the authors not directly penalizing for cost [LPQa] and the authors added "with penalty" results in the rebuttal. These results showed slightly lower accuracy while reducing the cost significantly below the target budget. In my view, the "no penalty" initial results were sufficient since the remaining budget was taken into account in the state vector (and the action space is simplified to monotonic increasing cost), and the total cost in the results was tightly $\le$ the budget, which satisfies Eq. 1. Further reducing the cost below the given budget was not an objective here (and I don't think it should have been), and I agree with the author's statement that this would naturally come with reduced accuracy. That said, I see the reviewer's point that more advanced reward and state design that takes into account the cost of different M+P instead of only sorting them could have been more interesting. The authors should include the total actual cost vs. allotted budget in the paper, and consider more advanced cost penalties.

The main weaknesses of the paper is that while it's focusing on a very practical setup, the suggested solution is not very practical. It assumes an in-domain set of labeled questions for training and applying a local network after each LLM response (and the cost of this network wasn't considered at all), a prior estimation of accuracy and cost for different model/prompt combinations, and that the test questions are given in order rather than all at once. At the end of the day, the simple "calibrated cascade" baseline that doesn't use all the bells and whistles of RL and policy learning performs about the same. Still, the formulation and setup could create an interesting baseline for future work on this field and could be of interest to the community. Unfortunately the authors didn't attach their code in the submission, but state that they will make it public. Having easy access for reproducing all baselines and the proposed method could facilitate more research on this topic